# Lookup multivariate Kolmogorov-Arnold Networks

**Sergey Pozdnyakov**[1]
sergey.pozdnyakov@epfl.ch

**Philippe Schwaller**[1,2]
philippe.schwaller@epfl.ch

[1] École Polytechnique Fédérale de Lausanne (EPFL)
[2] National Centre of Competence in Research (NCCR) Catalysis

## ABSTRACT

High-dimensional linear mappings, or linear layers, dominate both the parameter count and the computational cost of most modern deep-learning models. We introduce a general-purpose drop-in replacement, lookup multivariate Kolmogorov-Arnold Networks (lmKANs), which deliver a substantially better trade-off between capacity and inference cost. Our construction expresses a general high-dimensional mapping through trainable low-dimensional multivariate functions. These functions can carry dozens or hundreds of trainable parameters each, and yet it takes only a few multiplications to compute them because they are implemented as spline lookup tables. Empirically, lmKANs reduce inference FLOPs by up to $6.0\times$ while matching the flexibility of MLPs in general high-dimensional function approximation. In another feed-forward fully connected benchmark, on the tabular-like dataset of randomly displaced methane configurations, lmKANs enable more than $10\times$ higher H100 throughput at equal accuracy. Within the framework of Convolutional Neural Networks, lmKAN-based CNNs cut inference FLOPs at matched accuracy by $1.6$–$2.1\times$ and by $1.7\times$ on the CIFAR-10 and ImageNet-1k datasets, respectively. Our code, including dedicated CUDA kernels, is available online at https://github.com/schwallergroup/lmkan.

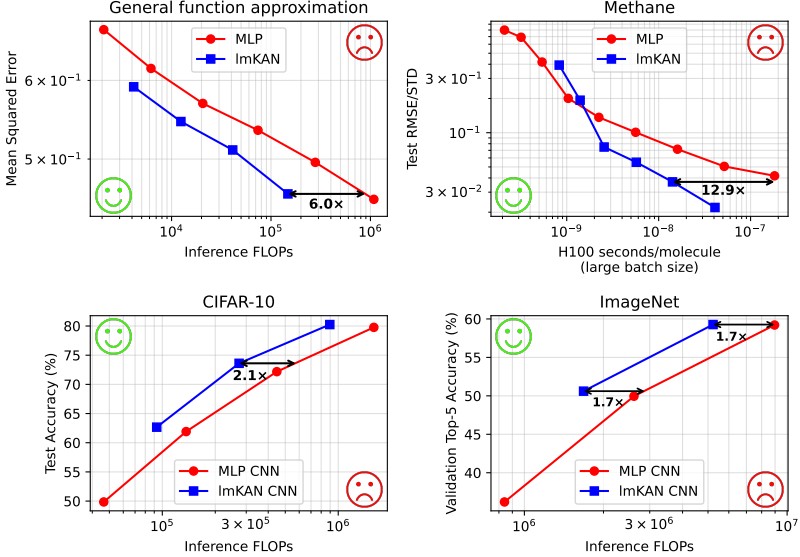

Figure 1: Performance summary. See Sec. 4 for details.

# 1 INTRODUCTION

With a sufficient amount of training data, the capabilities of deep-learning models systematically improve with the number of trainable parameters (Zhai et al., 2022; Kaplan et al., 2020). However, deploying very large models is challenging because of the associated inference cost.

In most models, high-dimensional linear mappings dominate both the parameter count and the computational cost. Standard multilayer perceptrons (MLPs) alternate linear layers with activations, and sometimes with a few other layers (Ioffe & Szegedy, 2015; Hinton et al., 2012). If $N$ is the width of a layer, the parameter count and inference cost of these linear mappings scale as $\mathcal{O}(N^2)$, whereas most other layers scale only as $\mathcal{O}(N)$. The same observation holds for many other architectures. Transformers (Vaswani et al., 2017), when applied to very long sequences, are one of the few notable exceptions because the cost of attention grows quadratically with the number of tokens. Even in that case, however, the cost of the linear mappings remains substantial, not to mention the potential use of fast approximations of attention (Choromanski et al., 2020).

The computational cost of a linear layer is proportional to the number of its parameters: at inference, each parameter induces one multiplication per input, where the input is a whole input object in the case of MLPs, a token in the case of Recurrent Neural Networks (Elman, 1990) and Transformers (Vaswani et al., 2017), a node or an edge in the case of Graph Neural Networks (Zhou et al., 2020), a patch of an image in the case of Convolutional Neural Networks (LeCun et al., 2002; Krizhevsky et al., 2012), and similarly for other architectures.

Spline lookup tables make it possible to do better than that. Consider, for example, a one-dimensional piecewise-linear function $f(x)$ on the interval from 0 to 1 with a uniform grid. On each interval, it is given as $f(x) = a[i] * x + b[i]$, where $i$ is the interval index. With $G$ intervals, the function has $2G$ parameters, out of which $G + 1$ are independent once continuity at the internal grid points is enforced. Yet the computational cost of evaluating such a function at any given point is $\mathcal{O}(1)$, not depending on $G$. The computational pipeline involves first determining the current grid interval as $i = \lfloor x * G \rfloor$, and then evaluating only one linear function as $f(x) = a[i] * x + b[i]$.

Kolmogorov-Arnold Networks (KANs) (Liu et al., 2024), designed as a general alternative to MLPs, are natural hosts for spline lookup tables as they construct a general high-dimensional mapping through a collection of trainable univariate functions. This design enables the introduction of numerous trainable parameters within each inner function without a commensurate increase in inference cost.

The main contributions of this work are the following:

- We propose lookup multivariate Kolmogorov-Arnold Networks (lmKANs) that are built upon multivariate low-dimensional functions instead of the univariate ones that standard KANs employ. We find that the multivariate version is better suited for a rich parametrization of inner functions in practice. When empirically comparing the 2D version of lmKANs against 1) 1D FastKAN (Li, 2024b) and 2) the 1D version of lmKANs, we find that the 2D lmKANs are more accurate and easier to train in both cases.

- We implement the inner functions as spline lookup tables. Ignoring a non-asymptotic $\mathcal{O}(N)$ term, the required inference FLOPs are only 2× those of a linear layer of the same shape, while the number of trainable parameters can be dozens or hundreds of times higher.

- We provide custom CUDA kernels that enable efficient inference of lmKANs on modern GPUs. When using the $8 \times 8$ tile size, on the H100 GPU, our implementation enables up to $\sim 88\times$ faster inference compared to a linear layer with the same number of trainable parameters.

- We empirically compare lmKANs and MLPs across diverse datasets, scales, and backbones, using varied experimental setups to obtain a comprehensive view of performance. Across these conditions, lmKANs are consistently Pareto-optimal with respect to inference FLOPs. The performance of lmKANs is summarized in Fig. 1.

## 2 Related work

The Kolmogorov-Arnold Representation Theorem (KART) (Kolmogorov; Arnold, 2009) states that a continuous function $f : [0, 1]^n \to \mathbb{R}$ can be represented as:

$$f(x_1, \ldots, x_n) = \sum_{q=1}^{2n+1} \Phi_q \left( \sum_{p=1}^{n} \phi_{q,p}(x_p) \right), \tag{1}$$

where $\phi_{q,p} : [0,1] \to \mathbb{R}$, and $\Phi_q : \mathbb{R} \to \mathbb{R}$ are continuous univariate functions. There has been a long debate (Girosi & Poggio, 1989; Schmidt-Hieber, 2021) on the usefulness of this theorem for machine learning because of the general non-smoothness and wild behavior of the inner functions. Nevertheless, it inspired the construction of Kolmogorov-Arnold Networks (Hecht-Nielsen, 1987; Igelnik & Parikh, 2003), whose layers are defined as $y_q = \sum_p f_{qp}(x_p)$, where $f_{qp}$ are trainable functions. Liu et al. (2024) introduced the modern version, which suggests stacking an arbitrarily large number of KAN layers and using an arbitrarily large number of neurons, similarly to MLPs. While Liu et al. (2024) illustrated strong performance of KANs, many test cases involve ground-truth functions with known, reasonably smooth KART or KART-like (matching KANs with more than one hidden layer and a larger number of neurons) closed-form representations. Subsequent works such as Yang & Wang (2024), Kundu et al. (2024), and Kashefi (2025) further reinforced the efficiency of KANs. On the contrary, Yu et al. (2024) found that KANs can fall short compared to MLPs for some tasks.

The idea of lookup-based $\mathcal{O}(1)$ computations of KAN univariate functions is sometimes briefly mentioned but rarely implemented in practice (Somvanshi et al., 2024; Ji et al., 2024), likely because of challenges associated with an efficient GPU implementation. Surprisingly, most of the research goes in the somewhat opposite direction. B-splines, piecewise polynomial basis functions used in the original KAN paper, have compact support and thus are well suited for $\mathcal{O}(1)$ inference. Subsequent works often replace them with dense basis functions, such as Chebyshev polynomials (SS et al., 2024) or Fourier harmonics (Xu et al., 2024). The case of FastKAN (Li, 2024b), which replaces sparse B-splines with similar-looking dense Gaussian radial basis functions exclusively for the sake of optimization, is especially notable.

A few works, such as Moradzadeh et al. (2024) and Huang et al. (2025), implement the lookup idea. Moradzadeh et al. (2024), however, predict B-spline coefficients using an MLP for the given grid points, which, thus, are not fully independent of each other. Additionally, it benchmarks inference cost against a naive implementation of KANs and not against MLPs, thus not claiming that their construction is Pareto optimal compared to MLPs, which are still state-of-the-art for general tasks. Huang et al. (2025) achieves remarkable efficiency on a small-scale problem from the original KAN paper by algorithm-hardware co-design using the TSMC 22 nm RRAM-ACIM chip. Poluektov & Polar (2025) and Polar & Poluektov (2021) employ piecewise linear parametrization suitable for $\mathcal{O}(1)$ inference but do not focus on inference efficiency. While Shtoff (2025) provides a pure PyTorch implementation of splined parametric curves and, in particular, the KAN architectures derived from them, it does not benefit from low-level CUDA optimizations as the present work.

In this work, we provide CUDA kernels for efficient inference and benchmark the introduced models against MLPs on general tasks where KART representations are not known in closed form and where there is no reason to believe that they are smoother than in other cases.

## 3 Lookup multivariate Kolmogorov-Arnold Networks

At first glance, given that the inference cost of spline lookup tables does not depend on the number of parameters, very expressive univariate functions with tens of thousands of trainable parameters each are an ideal match for the Kolmogorov-Arnold Representation Theorem. However, KANs rarely use more than a few dozen parameters per function in practice. The difference between a univariate function parametrized by tens of thousands of parameters and just a few dozen is the capability of the former to parametrize a very high frequency band. On the one hand, this expressivity is necessary for closely approximating the 'wild behavior' of KART inner functions, but on the other, it raises concerns about training stability and generalization. On the contrary, multivariate functions can "accommodate" a significantly larger number of parameters without spilling expressive power to exceedingly high frequency bands. For instance, a four-dimensional function with just 10 grid points

along each dimension has roughly the same number of trainable parameters as a univariate one with $\sim 10^4$ grid intervals.

A layer of a multivariate version of Kolmogorov-Arnold Networks with dimension $d$ defines the output as:

$$y_q = \sum_{p=0}^{N_{\text{in}}/d-1} f_{qp}(x_{dp}, x_{dp+1}..., x_{dp+d-1}),$$ (2)

where $f_{qp}$ are trainable d-dimensional functions and $N_{\text{in}}$ is the input dimensionality (assumed to be divisible by $d$). We implement CUDA kernels for the one- and two-dimensional cases. The motivation behind this choice is detailed in Section 3.3. An example of such a layer in the 2D case is depicted in Fig. 2. Similar to KANs, these layers do not need additional activations in between and can be stacked arbitrarily, substituting linear mapping-activation pairs in MLP-based backbones.

In Sec. 4.4, we empirically compare the two-dimensional version of lmKANs with both one-dimensional lmKAN and one-dimensional FastKAN. The outcomes of these numerical experiments reinforce the intuitive considerations given here and suggest that multidimensional building blocks can indeed be more effective hosts for a large number of parameters in a practical setup.

Additionally, it is worth noting that, if necessary, multivariate functions $f_{qp}$ can always fall back to sums of univariate ones, which would make the whole lmKAN fall back to standard KAN. Thus, the Kolmogorov-Arnold Representation Theorem is applicable also to our construction.

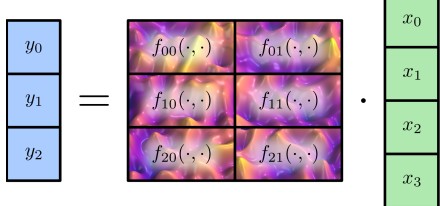

Figure 2: Schematic representation of a 2D lmKAN layer with 4 inputs and 3 outputs. This layer defines outputs as: $y_0 = f_{00}(x_0, x_1) + f_{01}(x_2, x_3)$, $y_1 = f_{10}(x_0, x_1) + f_{11}(x_2, x_3)$, and $y_2 = f_{20}(x_0, x_1) + f_{21}(x_2, x_3)$. The functions $f_{**}(\cdot, \cdot)$ are to be trained during fitting.

### 3.1 FUNCTION PARAMETRIZATION

During training, activations of neurons can evolve arbitrarily, making the use of grids defined on bounded regions challenging. Therefore, we designed an unbounded grid which is still regular enough to allow $\mathcal{O}(1)$ computations.

**Sigma grid**  The one-dimensional sigma grid, which is illustrated in the left panel of Fig. 3, is generated by any sigmoid-like function $\sigma(x)$. If the desired number of grid intervals is $G$, then the grid points are given as the intersections of $G-1$ equispaced percentile levels with $\sigma(x)$. Such a construction spans the entire real axis. The grid has the finest resolution near the origin, and becomes progressively coarser as $|x|$ increases. For a given $x$, the index of the corresponding grid interval can be computed as $i = \lfloor \sigma(x)G \rfloor$, which makes such a grid suitable for $\mathcal{O}(1)$ computations. For multivariate functions, we apply the same one-dimensional grid independently to each coordinate.

To balance occupancy across intervals, we precede each lmKAN layer with a batch normalization layer without affine parameters to keep the activations in a controlled range. The choice of the $\sigma(x)$ function was primarily motivated by computational efficiency. The exact definition and other details are available in Appendix B.

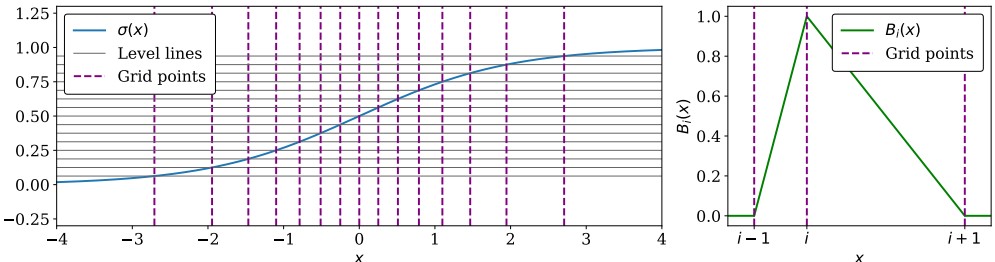

Figure 3: (Left) Construction of the sigma grid; (Right) Second-order B-spline.

**Compact basis functions** We use B-splines of second order built on top of the described grid as basis functions to parametrize the lmKAN inner functions. A one-dimensional second-order B-spline centered around grid point $i$ is given in the right panel of Fig. 3. It takes non-zero values on only the two adjacent grid intervals around the center grid point. If there are $G$ intervals, then we use $G + 1$ basis functions: $G - 1$ such B-splines centered around each inner grid point, and two linear functions on the leftmost and rightmost infinite intervals. Appendix B provides the definition of the edge-interval B-splines used in this work, along with other details. A two-dimensional B-spline is defined as $B_{i_1 i_2}(x_1, x_2) = B_{i_1}(x_1) B_{i_2}(x_2)$. All the functions defining a 2D lmKAN layer are parametrized as $f(x_1, x_2) = \sum_{i_1, i_2} p_{i_1 i_2} B_{i_1 i_2}(x_1, x_2)$, where $p_{i_1 i_2}$ are trainable coefficients.

With such a construction, there are $(G + 1)^2$ independent parameters for each 2D function, parametrized functions are bilinear on each 2D grid interval, all the functions are continuous for arbitrary $p_{i_1 i_2}$, and all but the edge coefficients $p_{i_1 i_2}$ have a simple interpretation as the value of the function on the corresponding grid point.

For any given point $(x_1, x_2)$, there are only four non-zero B-splines; thus, one needs to evaluate only four terms to compute $f(x_1, x_2)$, which forms the basis of $\mathcal{O}(1)$ computations. The full algorithm to compute a standalone two-dimensional function is given in Appendix B.1.

### 3.2 COMPUTATIONAL COST

Overall, the functional form of lmKANs involves computations of many low-dimensional functions with exactly the same arguments (those in the same column, see Fig. 2). When doing so, it is possible to reuse many of the intermediate values, such as indices of the grid intervals and B-splines. These intermediate values can be computed once for each pair of inputs, and then utilized to compute the value of a given function with just four multiply-add operations for the 2D case. Given that the total number of 2D functions in an lmKAN layer is $[N_{\text{in}}/2]N_{\text{out}}$, the total number of required multiply-add operations for the dominant, $\mathcal{O}(N^2)$, part is $4[N_{\text{in}}/2]N_{\text{out}} = 2N_{\text{in}}N_{\text{out}}$, just **2×** that of a linear layer of the same shape. Algorithm 3, given in Appendix C, pinpoints this estimation. Following the common practice (He et al., 2016), we estimate FLOPs as the number of fused multiply-adds of the main asymptotic term for both MLPs and lmKANs.

It is worth noting that the non-asymptotic $\mathcal{O}(N)$ term is not an additional cost relative to MLPs; it replaces the per-unit bias additions and activation evaluations that lmKANs do not require — operations that can be quite expensive when the activations are transcendental functions such as `tanh`.

### 3.3 PERSPECTIVE ON DIMENSIONS AND SPLINE ORDERS

All the constructions introduced so far straightforwardly generalize to a higher-dimensional case. Evaluation of a single $d$-dimensional function parametrized by d-dimensional second-order B-splines would take $2^d$ multiply-adds, which stem from the summation of B-spline contributions residing on all the corners of the corresponding d-dimensional hypercube. Given that the number of such d-dimensional functions would be $d$ times smaller than the number

of weights of a linear layer of the same shape, the inference FLOPs of the d-dimensional lmKAN layer would be $2^d/d$ of that of the linear layer with the same shape.

These slowdown factors are identical, $2\times$, for one- and two-dimensional lmKANs, and start to grow for higher dimensions. Thus, we chose the one- and two-dimensional versions for the implementation of CUDA kernels as the 2D extension of the standard univariate KAN comes essentially for free.

If B-splines of order $k$ are employed for parametrization instead of the second-order ones described so far, the inference cost becomes $k^d/d$ of that of an MLP. For more details, see the B-spline definition at De Boor & De Boor (1978), and how they are used in KANs (Liu et al., 2024). Increasing the B-spline order brings a few benefits, but they likely do not justify the associated increase in the computational cost.

The classical theorem about B-splines (De Boor, 1968) indicates that while the order of B-splines affects the convergence rate, any spline order is sufficient to approximate any function arbitrarily closely by increasing the resolution of the grid[1]. Given that the computational cost of spline look-up tables does not depend on the number of grid points, the grid resolution can be arbitrarily increased without any computational overhead at inference.

On top of that, increasing the B-spline order introduces additional smoothness to the model. Functions parametrized by B-splines of order $k$ belong to $C^{k-2}$, but in general not to $C^{k-1}$. The smoothness of second-order B-splines employed in this work matches that of ReLU (Glorot et al., 2011), one of the most popular and successful activation functions, which is also continuous, but not continuously differentiable. Thus, it is questionable if the additional smoothness available through higher orders $k$ is necessary and would justify the associated computational overhead.

### 3.4 CUDA KERNELS

We implement CUDA kernels for efficient inference of 1D and 2D lmKANs on modern GPUs. Our kernels use the classic shared memory tiling used in GEMM (Volkov & Demmel, 2008). In the following, all benchmarks run in full-precision `float32`.

In the 2D case, with a $16 \times 16$ tile, our implementation is $\sim 8\times$ slower than a dense linear layer with the same shape on an H100-SXM, irrespective of the grid resolution. The slowdown exceeds the $\sim 2\times$ FLOPs-based estimate from Sec. 3.2, primarily because of the less coherent memory-access pattern of algorithm 3. Additionally, dense matrix multiplication has been the cornerstone of many computational pipelines and, thus, has enjoyed decades of thorough optimization.

Finite shared memory capacity limits the number of grid intervals to $G \leq 20$ on H100. At this limit, an lmKAN layer holds $(20+1)^2/2 \approx 220\times$ more parameters than the linear baseline with the same shape. This means that an lmKAN layer delivers $\left[(20+1)^2/2\right]/8 \approx 27.5\times$ **faster** inference compared to a linear layer with the same number of trainable parameters.

Reducing the tile to $8 \times 8$ raises the slowdown to $\sim 9.5\times$ but lets us increase $G$ to 40, yielding $\approx 88.5\times$ **better per-parameter efficiency.**

## 4 RESULTS

We have demonstrated so far that lmKANs can have significantly better inference cost per trainable parameter compared to linear layers in terms of both FLOPs and wall-clock time on modern GPUs. The question is, however, whether this nominal efficiency translates to real-life performance. Do lmKANs indeed represent a better tradeoff between performance and inference cost?

In this section, we empirically compare the efficiency of lmKANs and MLPs across the following settings: (i) approximating general high-dimensional functions, (ii) on a tabular-like dataset of randomly displaced methane configurations, and (iii) within CNN frameworks

---

[1]This is applicable, though, to functions on a bounded domain.

evaluated on CIFAR-10 and ImageNet. Across all experiments, we use identical macro-architectural backbones for lmKANs and MLPs. Overall, to obtain a comprehensive picture of the performance, we prioritized the diversity of the setups over a very large scale or the architectural complexity of a particular backbone. We found that lmKANs are consistently inference FLOPs–accuracy Pareto-optimal, with the largest gains on the methane dataset. Finally, we compare lmKANs with FastKANs.

## 4.1 General function approximation

Our first experiment is set to measure the crude flexibility of lmKANs in approximating general high-dimensional functions, which we model by large teacher MLPs with fixed random weights. We define a ground-truth $\mathbb{R}^{32} \to \mathbb{R}^1$ function as an MLP with 32 input neurons, 10 hidden layers, each with 1024 neurons, and hyperbolic tangent activations.

We fit both MLP and lmKAN students to approximate this ground-truth function and compare their performance. We use the same fully connected backbone for both types of models with two hidden layers and varying hidden dimensions and grid resolutions $G$ (for more details, see Appendix G.2).

In our first experiment, we fixed the hidden dimension at 256 to investigate the performance of 1D and 2D lmKANs as a function of grid resolution $G$. The results, presented in Fig. 4 (left panel), show that performance does not improve indefinitely with higher resolution; instead, it follows a U-shaped curve. However, there is a notable distinction between the one- and two-dimensional cases: the 2D lmKAN saturates at a much larger number of trainable parameters per function and achieves significantly higher accuracy. Notably, the 2D lmKAN with optimal grid resolution performs comparably to an MLP that is roughly 16 times larger (with a 4× larger hidden dimension). Furthermore, the second and third panels confirm that inference cost—in terms of both FLOPs and H100 SXM wall-clock time—remains independent of the grid resolution $G$.

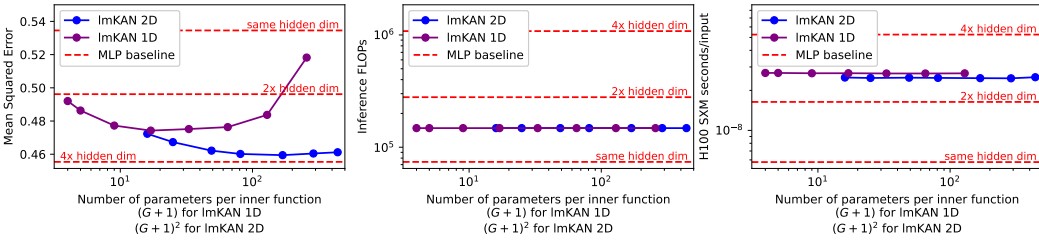

Figure 4: Performance of 1D and 2D lmKANs depending on the grid resolution.

In our second experiment, we fixed the grid resolution at its optimal value and varied the hidden dimensions for both MLPs and 2D lmKANs. As expected, the final Mean Squared Error (MSE) decreases monotonically with the hidden dimension, reflecting the increased flexibility of the models. However, larger hidden dimensions inevitably incur higher inference costs; the key question, therefore, is which architecture offers a better tradeoff between accuracy and efficiency. Our findings are summarized in Fig. 5. The left panel plots the final MSE against the hidden dimension, while the middle and right panels depict the Pareto fronts for MSE versus inference FLOPs and H100 wall-clock time, respectively.

In order to justify the claims that lmKANs are indeed more efficient, we converge the baseline MLP-based models very tightly here, and in all similar experiments in this manuscript. For the MLP baseline, we have two lines: one with a full training budget and one with only half of it. The close similarity between them demonstrates very tight convergence.

Fig. 5 clearly indicates that lmKANs are significantly more FLOP-efficient at the same accuracy level, up to **6×**, for the largest dimensions. Furthermore, lmKANs also appeared to be H100 wall-clock time optimal for all the scales, with a speedup factor of about 1.8× for the largest hidden dimension.

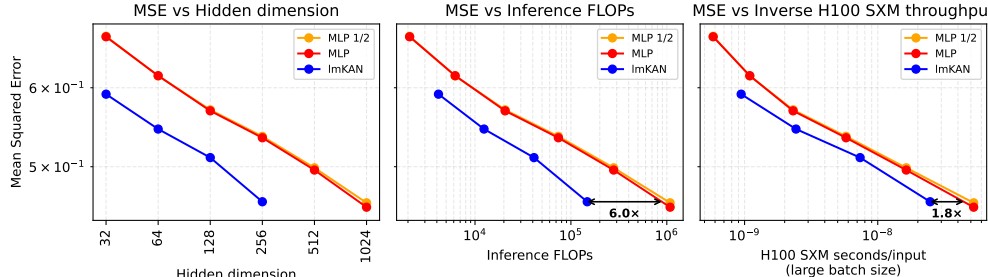

Figure 5: lmKAN vs MLP for general function approximation. The "MLP 1/2" line corresponds to the outcome of the fitting procedure with only half of the training steps compared to the "MLP" one.

Appendix G.2 contains an analogous experiment for an $\mathbb{R}^{32} \to \mathbb{R}^{32}$ function, ablation studies, and other details.

## 4.2 Randomly displaced methane configurations

Having demonstrated that lmKANs are Pareto-optimal when approximating a general function, we proceed to benchmark their efficiency on real data. We chose the tabular-like dataset of randomly displaced methane configurations for this comparison, as it is particularly suitable for this purpose (see Appendix G.4). The dataset consists of multiple off-equilibrium methane configurations, as illustrated in Fig. 6. The target is given by the corresponding quantum-mechanical energy (Turney et al., 2012; Kohn & Sham, 1965). Hydrogen atoms are placed around the carbon atom randomly, varying from instance to instance, which leads to different target energies.

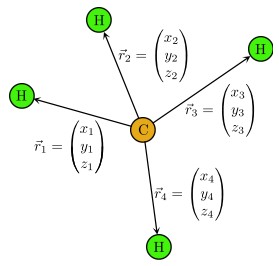

Figure 6: A methane configuration

We encode the geometry of each methane molecule by the Cartesian components of displacement vectors from the carbon atom to all the hydrogen atoms. Therefore, the input dimension of both the lmKAN and the MLP networks is 12. Otherwise, we use the same backbone with two hidden layers as in the previous experiment.

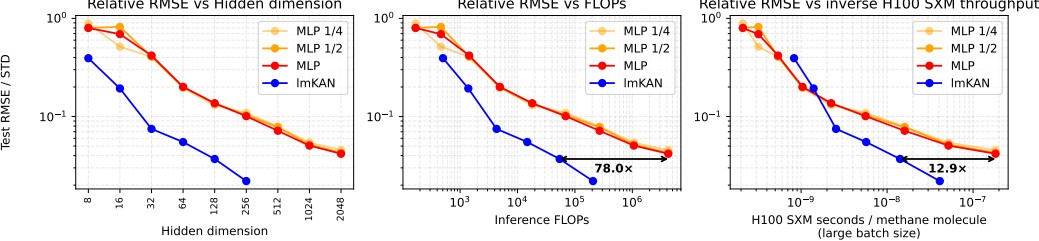

Figure 7: lmKAN vs MLP on the dataset of randomly displaced methane configurations. On the vertical axis, we plot the relative Root Mean Squared Error, which is given as test RMSE normalized by the standard deviation of the target in the dataset. The "MLP 1/2" and "MLP 1/4" curves correspond to outcomes of fitting procedures with half and a quarter of the training budget, respectively.

Our findings are given in Fig. 7. Similar to the previous experiment, lmKANs were found to be Pareto-optimal in terms of both formal FLOPs and H100 wall-clock time. Moreover, the

relative speedup is higher for this dataset: up to 78.0× reduced FLOPs, and 12.9× faster H100 inference at matched accuracy.

In Appendix G.4 we provide ablation studies, analogous experiments using representations other than the Cartesian components of the displacement vectors, and further details.

## 4.3 lmKAN-based Convolutional Neural Networks

In the introduction, we briefly outlined that high-dimensional linear mappings are the primary building blocks in most architectures, not only in feedforward fully connected neural networks. Convolutional Neural Networks (CNNs) are no exception.

A standard two-dimensional convolution with kernel size $k \times k$ is parametrized by a linear mapping $\mathbb{R}^{k^2 C_{in}} \to \mathbb{R}^{C_{out}}$, where $C_{in}$ and $C_{out}$ are the numbers of input and output channels, respectively. Since Kolmogorov-Arnold layers can be used as a general substitute for high-dimensional linear mappings, one can construct a KAN-based convolutional neural network well suited for image processing, as was done, e.g., in Bodner et al. (2024). In this section, we compare the performance of lmKAN- and MLP-based CNNs on the CIFAR-10 (Krizhevsky et al., 2009) and ImageNet (Deng et al., 2009) datasets.

For CIFAR-10, our backbone architecture consists of five $2 \times 2$ convolutions, each with stride 2, and two fully connected layers at the end. We use identical backbones for lmKAN and MLP CNNs, and exactly the same pool of augmentations (see more details in Appendix G.5). Similarly to the previous experiments, we vary the width of the neural networks, which is the number of filters in the case of convolutions, and the hidden dimension in the case of fully connected layers.

For ImageNet, we downsample the images to $81 \times 81$ pixels. Our backbone consists of four convolutional layers—each with a $3 \times 3$ kernel and a stride of 3—followed by two fully connected layers. In contrast to the CIFAR-10 experiment, we progressively increase the number of filters as the spatial resolution of the image decreases through the neural network.

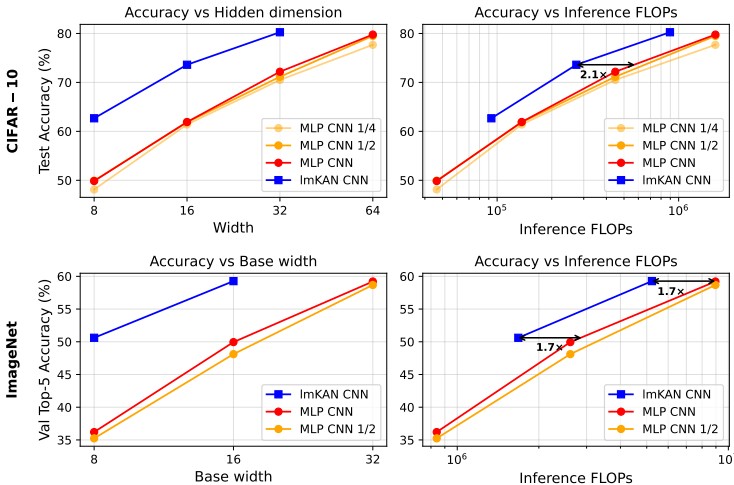

Figure 8: Comparison of the performance of standard MLP-based CNNs and lmKAN-based CNNs on the CIFAR-10 and ImageNet datasets. The "MLP CNN 1/2" line corresponds to the outcome of the fitting procedure with only half of the training steps compared to the "MLP CNN" one.

Our findings are illustrated in Fig. 8. Similarly to previous experiments, lmKAN-based CNNs were found to be more FLOPs efficient at the same accuracy level. The observed reduction in FLOPs is 1.6–2.1× for CIFAR-10, and 1.7× for ImageNet. We have not yet implemented dedicated CUDA kernels for efficient inference of lmKAN-based convolutions. Any type of convolution can be cast to a fully connected layer by the corresponding memory manipulations, which we employed during fitting.

### 4.4 COMPARISON WITH FASTKAN

In this section, we compare the performance of 2D lmKAN, 1D lmKAN, and FastKAN models. We use the training script[2] for the CIFAR-10 dataset available in the FastKAN GitHub repository (Li, 2024a) as the basis for the comparison. This script fits a fully connected FastKAN model with one hidden layer. We provide several modifications, particularly enabling the same augmentation pipeline we used in the previous section (Appendix G.6).

We fix the hidden dimensionality to 256 as in the original script for all the models, vary grid resolutions, and analyze the accuracy depending on the number of trainable parameters in each inner function. The result is given in Fig. 9.

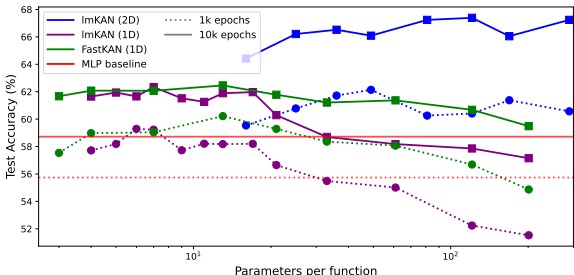

Figure 9: Comparison of the one- and two-dimensional lmKAN and FastKAN models within the fully connected framework on the CIFAR-10 dataset for fitting budgets of one and ten thousand epochs. The number of parameters per inner function is $(G+1)^2$ for 2D lmKAN, $(G+1)$ for 1D lmKAN, and $(\text{num\_grids} + 1)$ for FastKAN.

The first observation is that the performance of both FastKAN and 1D lmKAN models degrades for excessively fine grid resolutions. For the training budget of one thousand epochs, the final models become even less accurate than the MLP baseline. For ten thousand epochs, the effect is less pronounced, but it still takes place. This degradation is much less severe, if present at all, for the 2D lmKAN models—they are easier to fit even for a very rich parametrization of each inner function. Another distinct feature of Fig. 9 is that 2D lmKANs achieve notably better accuracy compared to both 1D lmKANs and FastKANs, even when the latter have a very rich parametrization of inner functions.

To sum up, these findings reinforce the intuitive considerations given in Sec. 3 and suggest that building blocks of multivariate trainable functions are indeed more effective.

## 5 SUMMARY

High-dimensional linear mappings are the cornerstone of modern deep learning, dominating both the parameter count and the computational cost in most models. We introduce lookup multivariate Kolmogorov-Arnold Networks (lmKANs) that offer a substantially better capacity–inference cost ratio. Across all experiments, lmKANs were found to be inference FLOPs Pareto-optimal. The efficiency gains are task-dependent: for general high-dimensional function approximation, modelled as a distillation from a large ground-truth teacher MLP with random weights, lmKANs achieved up to $6\times$ fewer FLOPs at matched accuracy. On randomly displaced methane configurations, efficiency improved by up to $78\times$. Within convolutional networks, the gains were smaller but still significant: $1.6$–$2.1\times$ on CIFAR-10 and $1.7\times$ on ImageNet.

Our CUDA kernels compete directly with highly optimized dense matrix multiplications—the backbone of many numerical pipelines for decades. Even so, the gains were sufficient to make lmKANs Pareto-optimal in H100 wall-clock time for both the general high-dimensional function approximation and the methane dataset, achieving a speedup of more than an order of magnitude in the latter case.

---

[2]https://github.com/ZiyaoLi/fast-kan/blob/master/examples/train_cifar10.py

## 6 REPRODUCIBILITY STATEMENT

The associated supplementary material contains not only the implementation of lmKANs, but also training scripts for the experiments presented in this manuscript, along with the corresponding settings as a collection of YAML files. These assets enable reproducibility of the numerical experiments presented in this paper.

## 7 ETHICS STATEMENT

This work adheres to the ethical standards expected at ICLR 2026. To the best of our knowledge, the proposed method is not expected to have adverse effects.

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

## A  LLM USAGE STATEMENT

We used LLMs for (i) polishing the text of the manuscript; (ii) finding related work and locating specific details in the corresponding papers; (iii) as a coding assistant during implementation of our codebase.

## B  MORE DETAILS ON FUNCTION PARAMETRIZATION

**The $\sigma(x)$ function**   Section 3.1 of the main text describes the construction of the sigma grid used in the parametrization of all the functions lmKAN consists of. It involves any sigmoid-like function $\sigma(x)$, and, as was mentioned in the main text, our choice, which is given in Eq. 3 and is illustrated in Fig. 10, was motivated by computational efficiency.

$$\sigma(x) = \begin{cases} 0.5\,e^x, & x \leq 0, \\ 1 - 0.5\,e^{-x}, & x > 0. \end{cases} \quad (3)$$

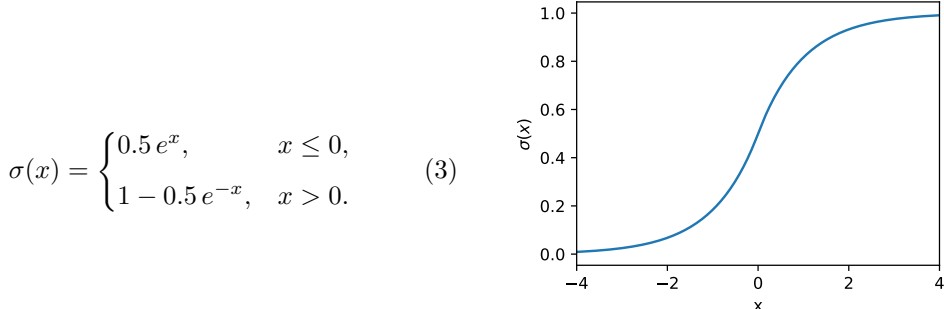

Figure 10: The $\sigma(x)$ function defined in Eq. 3.

This construction is cheap to compute because the computational pipeline consists of a single exponential call and a few arithmetic operations, as elaborated in algorithm 1.

---

**Algorithm 1** Evaluation of $\sigma(x)$ with a single exponential call

---

**Input:** $x \in \mathbb{R}$
**Output:** $\sigma(x)$
 1: $t \leftarrow \exp(-|x|)$         $\triangleright$ compute expensive exponential only once
 2: **if** $x > 0$ **then**
 3:    $\sigma(x) \leftarrow 1 - 0.5\,t$
 4: **else**
 5:    $\sigma(x) \leftarrow 0.5\,t$
 6: **end if**
 7: **return** $\sigma(x)$

---

**Edge cases**   Section 3.1 of the main text introduced sigma grids and the corresponding basis of second-order B-splines. For a grid with $G$ intervals and $G-1$ grid points, there are $G+1$ basis functions, out of which $G-1$ are given by second-order B-splines centered around all grid points, as illustrated in the right panel of Fig. 3 of the main text. The other two are given as linear functions on the leftmost and rightmost infinite intervals.

The second-order B-splines given in the right panel of Fig. 3 are defined to linearly increase from 0 to 1 from the left grid point to the central one and then linearly decrease from 1 to 0 from the central grid point to the right grid point. This construction is well-defined for all the inner grid points but requires additional definitions for the B-splines centered around the leftmost and rightmost grid points, as these do not have left and right neighboring grid points, respectively.

In order to define these edge B-splines, we introduce 'ghost' left and right grid points. The position of the left ghost point is given as $\boldsymbol{\mathcal{G}}[0] - (\boldsymbol{\mathcal{G}}[1] - \boldsymbol{\mathcal{G}}[0])$, where $\boldsymbol{\mathcal{G}}[0]$ and $\boldsymbol{\mathcal{G}}[1]$ are the positions of leftmost and second leftmost grid points respectively. The right ghost point is defined similarly. With such a notation of additional grid points, we can define edge B-splines similarly to all the others.

Finally, there are two linear basis functions on the leftmost and rightmost infinite intervals. We define them to be $0-$s in the leftmost and rightmost grid points, linearly increasing to 1-s at the left and right ghost points and continuing to left and right infinities with the same slope, respectively.

**Direct $\sigma$-grid vs. uniform grid after pre-normalization by $\sigma(x)$**   A natural question is how the proposed sigma grid construction differs from a simpler alternative that first maps the input via $x' = \sigma(x) \in (0, 1)$ and then fits a piecewise-linear function $g(x')$ on a uniform grid over $[0, 1]$.

In general, the resulting functions in the original $x$-domain, $f(x)$ and $f'(x) = g(\sigma(x))$, are not identical. Most importantly, $f'(x)$ is no longer piecewise linear within each $x$-grid interval. The most notable discrepancy appears in the tails: $f'(x)$ has horizontal (constant) asymptotes when $x \to \pm\infty$, whereas $f(x)$—by construction on the $\sigma$-grid—exhibits linear asymptotes.

Horizontal asymptotes entail vanishing gradients in the tails, a behavior widely implicated in training difficulties and one reason ReLU activations often outperform tanh (Glorot et al., 2011). For these reasons, we adopt the direct $\sigma$-grid parametrization in practice.

### B.1   STANDALONE TWO-DIMENSIONAL FUNCTION COMPUTATION

Algorithm 2 provides a recipe to compute a standalone two-dimensional function given our parametrization scheme. See discussion in Sec. 3.1 of the main text.

## C   COMPUTATIONAL COST

Algorithm 3 represents a computational pipeline to compute the forward pass of an entire 2D lmKAN layer.

---

**Algorithm 2** $\mathcal{O}(1)$ evaluation of a standalone 2D lmKAN function. Red lines indicate computations that can be reused when computing many 2D functions for the same arguments, while green lines indicate computations that have to be repeated for each 2D function.

---

**Input:** scalars $x_1, x_2 \in \mathbb{R}$; grid points $\mathcal{G}$; parameters $\mathbf{P} \in \mathbb{R}^{[G+1, G+1]}$
    $\mathbf{P}[i_1, i_2]$ stores the function value on the $(i_1, i_2)$-th grid point
**Output:** output $y \in \mathbb{R}$

---

1: **function Eval2D**$(x_1, x_2)$ ▷ Handling of edge-interval cases described above is omitted for brevity.
2:      $i_1 \leftarrow \lfloor \sigma(x_1) G \rfloor$
3:      $i_2 \leftarrow \lfloor \sigma(x_2) G \rfloor$
4:      $B_{i_1 i_2}(x_1, x_2) \leftarrow \dfrac{\mathcal{G}[i_1 + 1] - x_1}{\mathcal{G}[i_1 + 1] - \mathcal{G}[i_1]} \dfrac{\mathcal{G}[i_2 + 1] - x_2}{\mathcal{G}[i_2 + 1] - \mathcal{G}[i_2]}$
5:      $B_{i_1+1\, i_2}(x_1, x_2) \leftarrow \dfrac{x_1 - \mathcal{G}[i_1]}{\mathcal{G}[i_1 + 1] - \mathcal{G}[i_1]} \dfrac{\mathcal{G}[i_2 + 1] - x_2}{\mathcal{G}[i_2 + 1] - \mathcal{G}[i_2]}$
6:      $B_{i_1\, i_2+1}(x_1, x_2) \leftarrow \dfrac{\mathcal{G}[i_1 + 1] - x_1}{\mathcal{G}[i_1 + 1] - \mathcal{G}[i_1]} \dfrac{x_2 - \mathcal{G}[i_2]}{\mathcal{G}[i_2 + 1] - \mathcal{G}[i_2]}$
7:      $B_{i_1+1\, i_2+1}(x_1, x_2) \leftarrow \dfrac{x_1 - \mathcal{G}[i_1]}{\mathcal{G}[i_1 + 1] - \mathcal{G}[i_1]} \dfrac{x_2 - \mathcal{G}[i_2]}{\mathcal{G}[i_2 + 1] - \mathcal{G}[i_2]}$
8:      $y \leftarrow 0$
9:      $y \mathrel{+}= B_{i_1 i_2}(x_1, x_2)\, \mathbf{P}[i_1, i_2]$
10:      $y \mathrel{+}= B_{i_1+1\, i_2}(x_1, x_2)\, \mathbf{P}[i_1+1, i_2]$
11:      $y \mathrel{+}= B_{i_1\, i_2+1}(x_1, x_2)\, \mathbf{P}[i_1, i_2+1]$
12:      $y \mathrel{+}= B_{i_1+1\, i_2+1}(x_1, x_2)\, \mathbf{P}[i_1+1, i_2+1]$
13:      **return** $y$
14: **end function**

---

As the algorithm shows, the preamble part contributes only to an asymptotically insignificant $\mathcal{O}(N)$ term, where $N$ is the input ($N_{\text{in}}$) or output ($N_{\text{out}}$) dimension. Given that the total number of 2D functions in an lmKAN layer is $[N_{\text{in}}/2]N_{\text{out}}$, the total number of required multiply-add operations for the dominant, $\mathcal{O}(N^2)$, part is $4[N_{\text{in}}/2]N_{\text{out}} = 2N_{\text{in}}N_{\text{out}}$, just **2×** that of a linear layer of the same shape.

## D    Hessian regularization

Direct fitting of splined functions with a fine grid imposes additional challenges related to generalization. The problem is illustrated in Fig. 11a for the simple case of fitting a standalone one-dimensional function parametrized by second-order B-splines on a uniform grid on $[0, 1]$ with $G = 40$ intervals, $f(x) = \sum_i p_i B_i(x)$.

Ground truth is an exact parabola, and the training set consists of 5 points, which are illustrated on the plot. Since the number of grid points is large, the model has enough flexibility to reproduce the training set exactly, but generalization is quite off. With such a fine grid, only a few B-splines, marked as bold on the bottom panel of Fig. 11a, take non-zero values for the training points. Thus, only the coefficients $p_i$ associated with these active B-splines receive non-zero gradient during training, while all the others have no incentive to evolve from the random values assigned at initialization. Standard L2 regularization is not much better, as it simply pushes all non-active coefficients $p_i$ to zero, which also results in a non-meaningful approximation after training.

When dealing with a similar problem, Xie et al. (2023) employed off-diagonal regularization based on a finite-difference scheme for the second derivative. One can put regularization terms as $\lambda \sum_i (p_i - p_{i+1})^2$ for first derivative, $\lambda \sum_i (p_i - 2p_{i+1} + p_{i+2})^2$ for second, and so on. Such regularization schemes result in meaningful approximations after training, as illustrated in Fig. 11b.

---

**Algorithm 3** Forward pass of a 2D lmKAN layer.

---

**Input:** input vector $\mathbf{x} \in \mathbb{R}^{N_{\text{in}}}$, parameter tensor $\mathbf{P} \in \mathbb{R}^{\left[G+1,\, G+1,\, N_{\text{in}}/2,\, N_{\text{out}}\right]}$.
    $\mathbf{P}[i_1,\, i_2,\, input\_index,\, output\_index]$ is the value of $f_{input\_index,\, output\_index}$ at the $i_1-,\, i_2-$th grid point.
**Output:** output vector $\mathbf{y} \in \mathbb{R}^{N_{\text{out}}}$
 1: $\mathbf{y} \leftarrow \mathbf{0}$
 2: **for** $input\_index = 0$ **to** $N_{\text{in}}/2 - 1$ **do**
 3:     $i_1, i_2, B_{i_1 i_2}(x_1, x_2), B_{i_1+1\, i_2}(x_1, x_2), B_{i_1\, i_2+1}(x_1, x_2), B_{i_1+1\, i_2+1}(x_1, x_2) \leftarrow$
              **Preamble**$\left(\mathbf{x}[2 \cdot input\_index], \mathbf{x}[2 \cdot input\_index + 1]\right)$
 4:     **for** $output\_index = 0$ **to** $N_{\text{out}} - 1$ **do**
 5:         $\mathbf{y}[output\_index] \mathrel{+}= B_{i_1\, i_2}(x_1, x_2)\, \mathbf{P}[i_1,\, i_2,\, input\_index,\, output\_index]$
 6:         $\mathbf{y}[output\_index] \mathrel{+}= B_{i_1+1\, i_2}(x_1, x_2)\, \mathbf{P}[i_1{+}1,\, i_2,\, input\_index,\, output\_index]$
 7:         $\mathbf{y}[output\_index] \mathrel{+}= B_{i_1\, i_2+1}(x_1, x_2)\, \mathbf{P}[i_1,\, i_2{+}1,\, input\_index,\, output\_index]$
 8:         $\mathbf{y}[output\_index] \mathrel{+}= B_{i_1+1\, i_2+1}(x_1, x_2)\, \mathbf{P}[i_1{+}1,\, i_2{+}1,\, input\_index,\, output\_index]$
 9:     **end for**
10: **end for**

11: **function Preamble**$(x_1, x_2)$     ▷ Handling of edge-interval cases is omitted for brevity.
    See Appendix B for more details.
**Input:** $x_1, x_2$; precomputed grid points $\boldsymbol{\mathcal{G}}$;
    precomputed inverse areas $\mathbf{A}_{\text{inv}}[i_1, i_2] = \left[(\boldsymbol{\mathcal{G}}[i_1 + 1] - \boldsymbol{\mathcal{G}}[i_1])(\boldsymbol{\mathcal{G}}[i_2 + 1] - \boldsymbol{\mathcal{G}}[i_2])\right]^{-1}$
**Output:** indices $i_1, i_2$ and
    B-splines $B_{i_1\, i_2}(x_1, x_2), B_{i_1+1\, i_2}(x_1, x_2), B_{i_1\, i_2+1}(x_1, x_2), B_{i_1+1\, i_2+1}(x_1, x_2)$
12:     $i_1 \leftarrow \left\lfloor \sigma(x_1)\, G \right\rfloor$
13:     $i_2 \leftarrow \left\lfloor \sigma(x_2)\, G \right\rfloor$
14:     $B_{i_1\, i_2}(x_1, x_2) \leftarrow (\boldsymbol{\mathcal{G}}[i_1 + 1] - x_1)(\boldsymbol{\mathcal{G}}[i_2 + 1] - x_2)\, \mathbf{A}_{\text{inv}}[i_1, i_2]$
15:     $B_{i_1+1\, i_2}(x_1, x_2) \leftarrow (x_1 - \boldsymbol{\mathcal{G}}[i_1])(\boldsymbol{\mathcal{G}}[i_2 + 1] - x_2)\, \mathbf{A}_{\text{inv}}[i_1, i_2]$
16:     $B_{i_1\, i_2+1}(x_1, x_2) \leftarrow (\boldsymbol{\mathcal{G}}[i_1 + 1] - x_1)(x_2 - \boldsymbol{\mathcal{G}}[i_2])\, \mathbf{A}_{\text{inv}}[i_1, i_2]$
17:     $B_{i_1+1\, i_2+1}(x_1, x_2) \leftarrow (x_1 - \boldsymbol{\mathcal{G}}[i_1])(x_2 - \boldsymbol{\mathcal{G}}[i_2])\, \mathbf{A}_{\text{inv}}[i_1, i_2]$
18:     **return** $i_1, i_2, B_{i_1\, i_2}(x_1, x_2), B_{i_1+1\, i_2}(x_1, x_2), B_{i_1\, i_2+1}(x_1, x_2), B_{i_1+1\, i_2+1}(x_1, x_2)$
19: **end function**

---

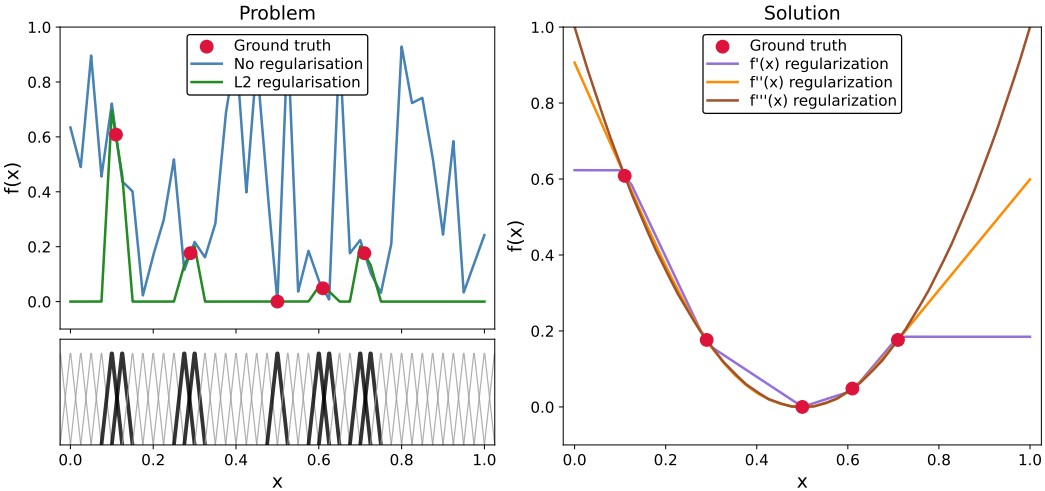

Figure 11: Generalization pitfall.

We implemented CUDA kernels for 2D lmKANs, which express general high-dimensional mapping in terms of building blocks of two-dimensional functions. For 2D functions, we use off-diagonal regularization based on the squared Frobenius norm of the Hessian, a

rotationally invariant measure of curvature in any direction, which is not to be confused with the Laplacian. Furthermore, our finite-difference schemes take into account that the grids, introduced in Sec. 3.1, are not uniform. The exact formulas are given below.

The Hessian of a function is zero if and only if the function is linear. Therefore, the use of a very strong Hessian-based regularization leads to linearization of the trained functions, enforcing them to converge to $f(x_1, x_2) = ax_1 + bx_2 + c$. This makes the whole lmKAN equivalent to an MLP of the same shape, modulo training dynamics. In other words, the Hessian regularization coefficient $\lambda$ can be used to smoothly adjust the lmKAN behavior between fully unconstrained lmKAN and MLP extremes. This observation that lmKANs with heavy Hessian regularization match non-regularized MLPs suggests that one should use a combination of the proposed regularization scheme with standard ones, such as L2 or dropout (Srivastava et al., 2014), for the best results.

### D.1 Detailed formulation of the Hessian regularization

We use the following finite-differences approximation for the second derivative with respect to $x_1$:

$$\left.\frac{\partial^2 f}{\partial x_1^2}\right|_{(x_1, x_2)} \approx \frac{2\big(h_\ell\, f(x_1 + h_r,\, x_2) - (h_\ell + h_r)\, f(x_1,\, x_2) + h_r\, f(x_1 - h_\ell,\, x_2)\big)}{h_\ell\, h_r\, (h_\ell + h_r)}, \tag{4}$$

where $h_l$ is the spacing between left and central grid points, while $h_r$ is the spacing between central and right grid points. The corresponding expression in terms of the coefficients $p_{i,j}$ is given as:

$$D_{x_1, x_1; i, j} = \frac{2\Big(h_i\, p_{i+1,j}\ -\ (h_i + h_{i+1})\, p_{i,j}\ +\ h_{i+1}\, p_{i-1,j}\Big)}{h_i\, h_{i+1}\, (h_i + h_{i+1})} \tag{5}$$

For the second derivative with respect to $x_2$ we use an analogous expression:

$$\left.\frac{\partial^2 f}{\partial x_2^2}\right|_{(x_1, x_2)} \approx \frac{2\big(h_b\, f(x_1,\, x_2 + h_u) - (h_b + h_u)\, f(x_1,\, x_2) + h_u\, f(x_1,\, x_2 - h_b)\big)}{h_b\, h_u\, (h_b + h_u)}, \tag{6}$$

where $h_u$ and $h_b$ are upper and bottom spacings, respectively.

$$D_{x_2, x_2; i, j} = \frac{2\Big(h_j\, p_{i,j+1}\ -\ (h_j + h_{j+1})\, p_{i,j}\ +\ h_{j+1}\, p_{i,j-1}\Big)}{h_j\, h_{j+1}\, (h_j + h_{j+1})} \tag{7}$$

For the mixed derivative our finite-differences scheme is the following:

$$\left.\frac{\partial^2 f}{\partial x_1\, \partial x_2}\right|_{(x_1, x_2)} \approx$$
$$\frac{f(x_1 + h_r,\, x_2 + h_p) - f(x_1 + h_r,\, x_2 - h_b) - f(x_1 - h_l,\, x_2 + h_p) + f(x_1 - h_l,\, x_2 - h_b)}{(h_r + h_l)\,(h_p + h_b)} \tag{8}$$

$$D_{x_1, x_2; i, j} = \frac{p_{i+1,j+1} - p_{i+1,j-1} - p_{i-1,j+1} + p_{i-1,j-1}}{(h_i + h_{i+1})\,(h_j + h_{j+1})} \tag{9}$$

The final regularization term is the following:

$$H_{i,j} = D_{x_1, x_1; i, j}^2 + 2D_{x_1, x_2; i, j}^2 + D_{x_2, x_2; i, j}^2. \tag{10}$$

In order to compute the total regularization term for the whole model, we 1) average $H_{i,j}$ across all the grid points and 2) sum these values across all the 2D functions within all the lmKAN layers in the model.

# E CUDA KERNELS

The performance of our CUDA kernels with 16x16 tile size in the limit of large dimensions is summarized in Fig. 12. All the lmKAN curves are computed with the largest number of grid intervals $G = 20$ available for the 16x16 tile size. We compare the inference efficiency of lmKAN and linear layers. On the left panel, we normalize time by the shape of the layers. Fig. 12 illustrates a clear convergence of these normalized times to the same value for all the dimensions. In the limit of large batch sizes, the forward pass of an lmKAN layer is ~8× slower compared to a linear layer with the same shape. At the same time, an lmKAN layer contains a significantly larger number of parameters than a linear layer of the same shape. Thus, inference time per parameter is significantly better for lmKAN layers, about 27 times, as illustrated on the right panel of Fig. 12.

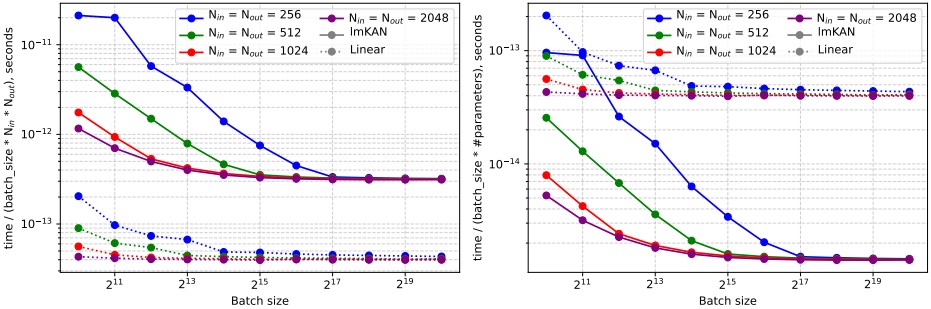

Figure 12: The performance of our CUDA kernels on the H100 SXM GPU in comparison with the linear layer in the limit of large dimensions. Left panel - time normalized by shape. Right panel - time normalized by the number of parameters.

Fig. 13 is an analogous illustration but for small dimensions - 16 and 32. Our CUDA kernels are better adjusted for such small dimensions, and thus, relative performance compared to linear layers is even higher in this case.

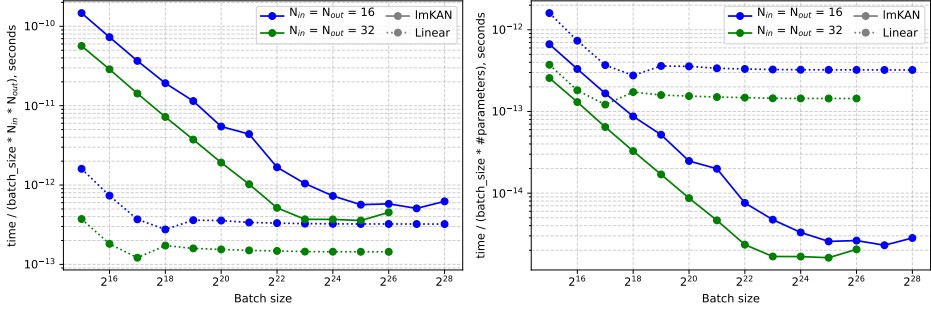

Figure 13: The performance of our CUDA kernels on the H100 SXM GPU in comparison with the linear layer for small dimensions. Left panel - time normalized by shape. Right panel - time normalized by the number of parameters.

Finally, Fig. 14 illustrates the inference efficiency depending on the number of grid intervals $G$, which control the number of parameters. The time indeed does not depend on $G$ in the large batch size limit.

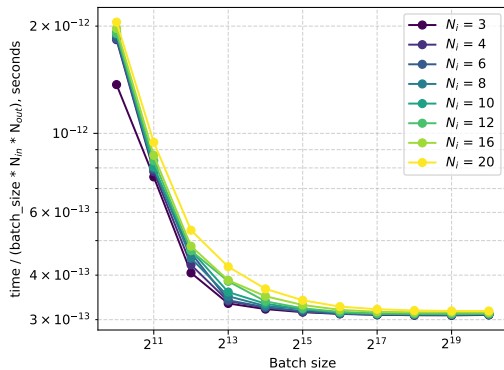

Figure 14: Inference efficiency of an lmKAN layer depending on the number of grid intervals $G$.

## F    PRECONDITIONING AND FITTING SCHEME

The first thing we attempted upon implementing the CUDA kernels was to fit a model with the highest grid resolution, $G = 40$, supported for the $8 \times 8$ tile on the H100 GPU. In this setup, each 2D function had as many as $41^2 = 1681$ trainable parameters. We found that the training was unstable, so we designed a preconditioning and multi-stage fitting pipeline to stabilize it. We employed this pipeline consistently for all our experiments.

The subsequent evidence revealed that lmKANs (similarly to KANs, as Sec. 4.4 illustrates) are progressively harder to fit as grid resolution increases. In other words, our very first experiment was the most challenging one. At more moderate grid resolutions, preconditioning measures can likely be simplified, if not omitted altogether. Specifically, we think that a fitting scheme omitting additional preconditioning terms, but preserving the Hessian regularization decay phase, which is described in the following, could be effective. With that, below is the description of the current pipeline.

### F.1    PRECONDITIONING

We precondition lmKAN layers by adding linear terms into the overall functional form. We use one of the following:

$$y = \gamma \, \mathrm{lmKAN}(x) + \mathrm{ReLU}\big(\mathrm{Linear}(x)\big) \tag{11a}$$

$$y = \gamma \, \mathrm{lmKAN}(x) + \mathrm{Linear}\big(\mathrm{ReLU}(x)\big) \tag{11b}$$

where the lmKAN weight, $\gamma$, is first set to 0 and then is gradually increased in our multistaged fitting procedure described later. In the case of ReLU-last preconditioning of Eq. 11a, we insert ReLU into all the layers except the last one; for ReLU-first preconditioning of Eq. 11b, we insert ReLU into all the layers except the first one. Therefore, at the beginning, when the lmKAN weight is zero, the model is equivalent to a pure MLP-based one for both types of preconditioning.

A merit of ReLU-first preconditioning is that during inference the whole Eq. 11b can be absorbed into a single lmKAN layer whenever the number of grid intervals $G$ is even, that is, when the origin is one of the grid points, see more details in Appendix G.1. Thus, this type of preconditioning does not increase inference cost in any way. This is an advantage over the original KAN preconditioning scheme (Liu et al., 2024), which requires the additional computation of the computationally expensive transcendental function SiLU for each edge at inference.

Because of the possibility of such an absorption, the total inference FLOPs of a ReLU-first preconditioned lmKAN layer are $2\times$ those of a linear layer of the same shape, while for the ReLU-last preconditioning, the slowdown factor is $3\times$, taking into account the linear branch.

## F.2 Fitting procedure

Our fitting scheme consists of several phases:

**Phase I - pure MLP:** $\gamma$ is set to 0, so the whole architecture is operating in pure MLP mode. This part is typically very short.

**Phase II - turning on lmKAN:** $\gamma$ is linearly (over time) increased from 0 to 0.3. After that, there is some part with the constant $\gamma = 0.3$. At this phase, we use very strong ($=$ with very high coefficient $\lambda$) Hessian regularization introduced in Appendix D. For all the subsequent phases, lmKAN weight $\gamma$ is fixed at 0.3. The pipeline is also stable if increasing $\gamma$ to 1.0, but in a few, though non-systematic, experiments, we found that using 0.3 value leads to slightly better final accuracy.

**Phase III - Hessian regularization decay:** At this phase, we gradually decay the strength of the Hessian regularization $\lambda$ from the initial very high value to the target value if this regularization is intended to be utilized in this fitting procedure and to nearly zero otherwise.

**Phase IV - Main lmKAN fitting part:** In this final phase, we keep Hessian regularization constant at the value reached in the previous phase. The model is fit with the given learning rate schedule. In the experiments in this work, we use a constant learning rate for the most part of this phase, and step or exponential learning rate decay at the end.

An example of the described fitting procedure for one of the training runs we did for numerical experiments described in Sec. 4.1 of the main text and in the Appendix G.2 is given in Fig. 15.

## F.3 Comparison of the preconditioning schemes

We fitted lmKAN models with both types of preconditioning for the CIFAR-10 dataset and for general function approximation. The results are given in Fig. 16 and Fig. 17. Overall, the ReLU-last type of preconditioning appeared to lead to slightly more accurate models, but this small gain in accuracy does not justify additional computational cost.

When designing some of our experiments we did not know this yet. Therefore, some of them use the ReLU-last type of preconditioning.

We use the ReLU-last type of preconditioning for Figures 21, 22, 23, and 24. We use the ReLU-first type of preconditioning for Figures 5, 8, 9, 26, and 27.

In other words, the performance of lmKANs on the methane datasets can likely be further improved by switching from the ReLU-last type of preconditioning to the ReLU-first one. However, since the observed gains in efficiency are already more than an order of magnitude in terms of the H100 wall-clock time, we left this for future work.

## G Experiments

### G.1 General details about the benchmarking protocols

Within the scope of this work, we primarily focus on the saturated throughput in the limit of large batch sizes. Thus, we benchmark all the models for progressively large batch sizes until reaching saturation. All the models are benchmarked with **10** warm-up dry runs, and **20** timed runs. Overall, we tried to optimize each model as much as possible while staying within the limits of full precision `float32` data type.

MLPs employed in this work consist of three types of layers - linear ones, ReLU activations, and batch normalizations. At inference, batch normalizations simply perform elementwise linear transformation and thus can be absorbed into the weights of linear layers. We perform this operation manually and, on top of that, compile the model with the `torch_tensorrt`

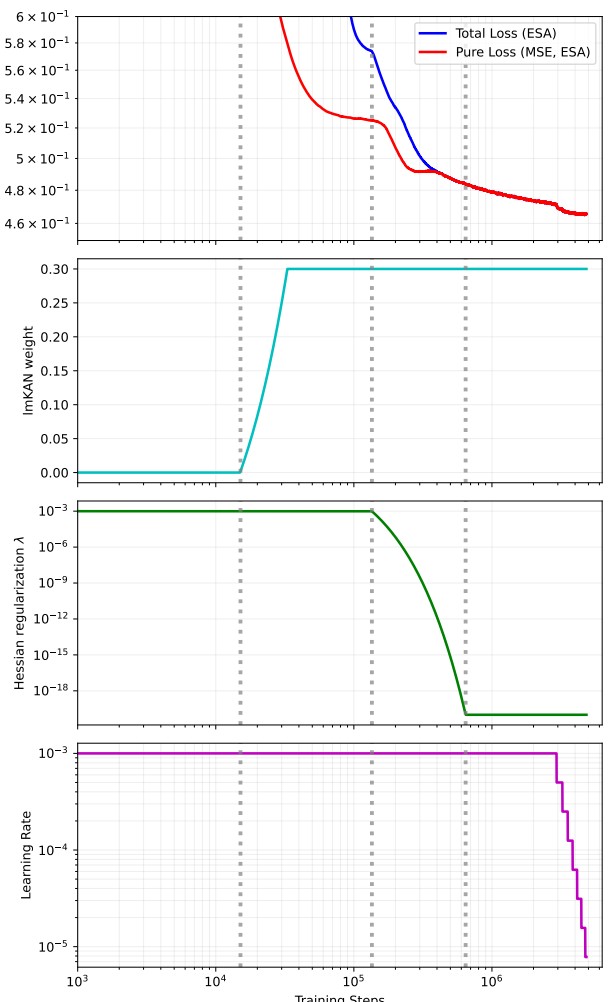

Figure 15: The multi-stage fitting procedure we use. Total loss indicates the full loss, including the Hessian regularization term. Pure loss is only the MSE part. For clarity, we plot exponential moving averages of losses. Note that the horizontal scale is logarithmic. If it is linear, the first couple of phases are hard to discern as they are very short. The fourth phase takes most of the training budget. This training run corresponds to the unregularized lmKAN, where Hessian regularization is turned on only at the beginning of the fitting to ensure stability. At the end of phase III, it reaches a nearly zero value, which, in this case, is $10^{-20}$.

backend (with `tf32` disabled to ensure full precision `float32`). We use the same compilation strategy for FastKANs.

For lmKANs, when using ReLU-first preconditioning (see more details in Appendix F), we absorb the entire expression in Eq. 11b into the weights of the lmKAN layer. This modification requires updating the lmKAN 2D functions as $f(x_1, x_2) \leftarrow \gamma f(x_1, x_2) + w_1 \text{ReLU}(x_1) + w_2 \text{ReLU}(x_2)$. Our construction allows for doing this absorption exactly whenever the origin is one of the grid points, which, in turn, is the case when the number of grid intervals $G$ is even. On top of that, batch normalizations are absorbed similarly to MLPs. We do not compile lmKAN models.

For lmKAN models with the ReLU-last preconditioning we absorb only the lmKAN weight $\gamma$.

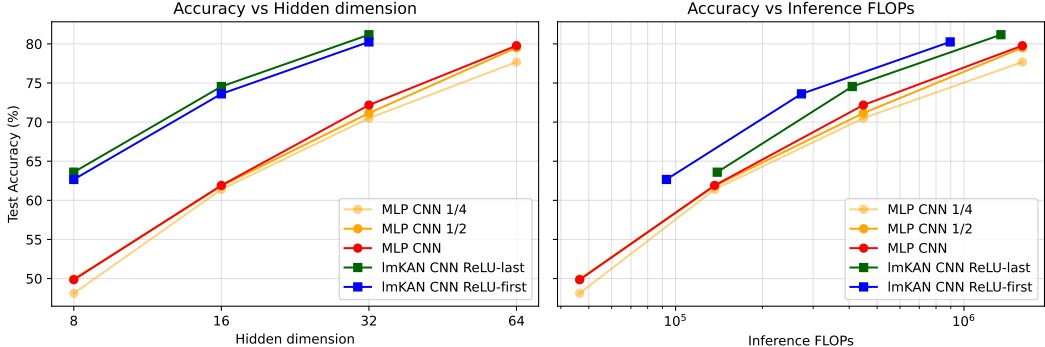

Figure 16: Comparison of the preconditioning schemes when fitting lmKANs on the CIFAR-10 dataset.

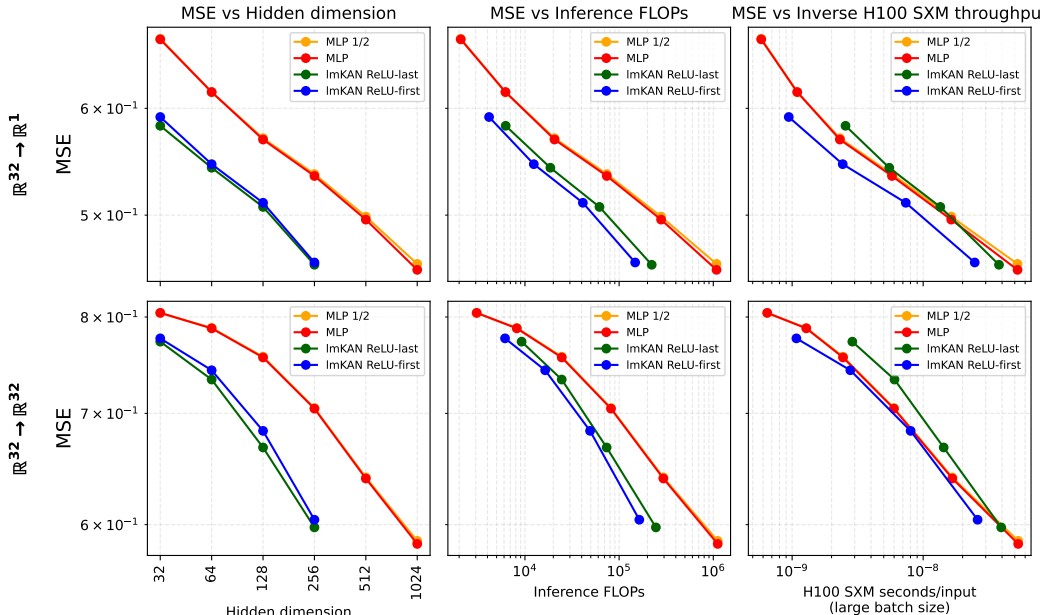

Figure 17: Comparison of the preconditioning schemes when fitting lmKANs within general function approximation setup.

### G.2 General Function Approximation

Both MLPs and lmKANs use batch normalizations. We set `affine=True` for MLPs as it is the standard choice, and `affine=False` for lmKANs in accordance with sigma grids introduced in Sec. 3.1. MLPs use ReLU activations, while lmKANs do not require any additional activation functions. We use $G = 12$ for all the lmKAN models, as this was the optimal value found in the ablation study described below. Pseudocode for both models is available in Fig. 18. Both students have two hidden layers, which is one more than both Cybenko (Cybenko, 1989) (the one for MLPs) and Kolmogorov-Arnold universal approximation theorems require. This setup, however, is more realistic, as MLPs with exactly one hidden layer are rarely used in practice.

<div>

**MLP student**

```
Linear(input_dim → hidden_dim)
BatchNorm1d(hidden_dim, affine=True)
ReLU()
Linear(hidden_dim → hidden_dim)
BatchNorm1d(hidden_dim, affine=True)
ReLU()
Linear(hidden_dim → output_dim)
```

</div>

<div>

**lmKAN student**

```
lmKANLayer(input_dim → hidden_dim)
BatchNorm1d(hidden_dim, affine=False)

lmKANLayer(hidden_dim → hidden_dim)
BatchNorm1d(hidden_dim, affine=False)

lmKANLayer(hidden_dim → output_dim)
```

</div>

Figure 18: Pseudo code for MLP and lmKAN students.

The performance of lmKANs when approximating a $\mathbb{R}^{32} \to \mathbb{R}^{32}$ function generated similarly as the $\mathbb{R}^{32} \to \mathbb{R}^1$ described in the main text is given in Fig. 19.

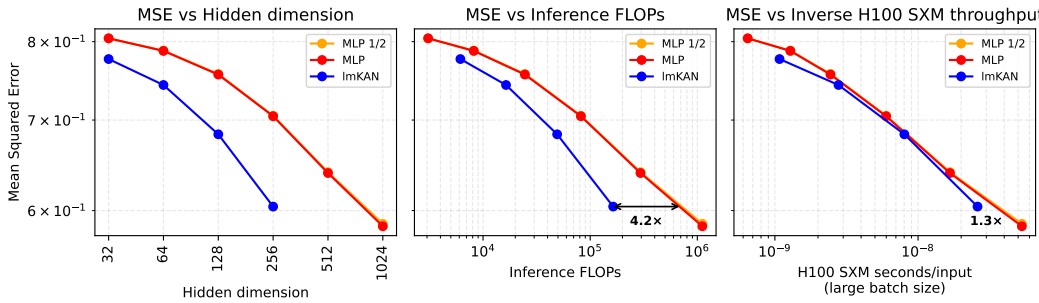

Figure 19: lmKAN vs MLP for general, $\mathbb{R}^{32} \to \mathbb{R}^{32}$, function approximation. The "MLP 1/2" line corresponds to the outcome of the fitting procedure with only half of the training steps compared to the "MLP" one.

There is a trend that the relative performance of lmKANs improves with the scale. It is clearly seen on the MSE vs FLOPs panel. On the MSE vs H100 wall-clock time panel, it is first masked by the non-homogeneous efficiency of the code, but next still reveals itself for the largest hidden dimensions.

We investigate the effect of the chosen number of grid intervals $G$ on the resulting lmKAN accuracy when approximating the $\mathbb{R}^{32} \to \mathbb{R}^1$ function with hidden_dim = 256. The result is given in Fig. 21.

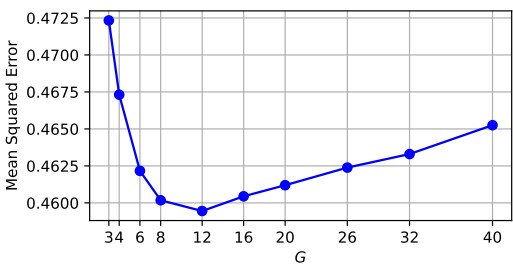

Figure 20: Final MSE vs $G$ for the hidden_dim=256 lmKAN model.

### G.3 Plot with shared axis for the number of parameters per each inner function

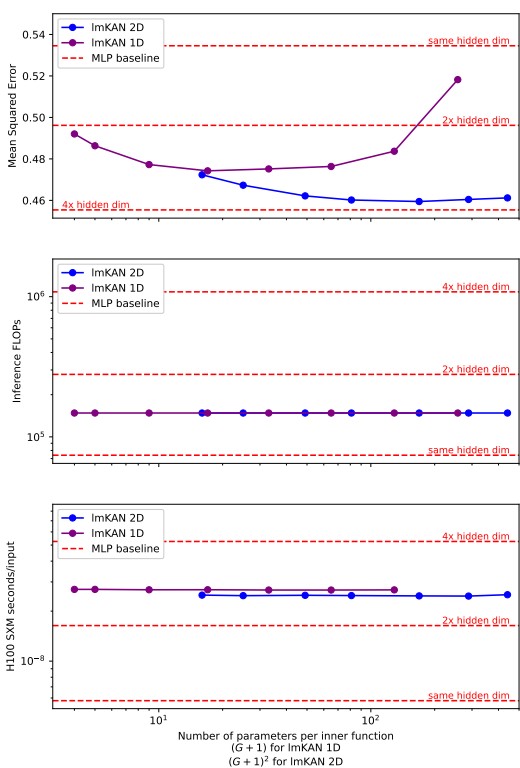

Figure 21: The same plot as Fig. 4 of the main text, but in vertical format and with the shared x axis.

### G.4 Methane

Tabular datasets are the natural choice for feedforward fully connected neural networks. Popular tabular datasets, such as the Titanic (Kaggle) or housing prices (Harrison Jr & Rubinfeld, 1978), however, are not particularly convenient for this purpose. First, they are typically stochastic in nature - for instance, while it is possible to improve a guess on the survival based on the data available for the Titanic dataset, it is impossible to say for sure. Thus, even an arbitrarily large model fitted on arbitrarily many data points would have a non-zero limitation on the accuracy. In other words, the performance of a model translates into an error metric not so directly, making the comparisons between different models less illustrative. Second, these datasets are typically relatively small, making it challenging to sweep across a wide range of model scales to obtain a comprehensive picture of performance.

Machine learning models fit on such datasets belong to the class of so-called machine learning interatomic potentials (Behler & Parrinello, 2007; Bartók et al., 2010). This dataset is sufficiently large for the comparisons, containing more than seven million configurations. Additionally, this dataset is deterministic - the geometry of the corresponding methane configuration completely determines the target (formally, there can be a stochastic term due to the lack of complete convergence of ab initio computations for the quantum-mechanical energy, but it is negligible in practice).

Table 1: Summary of methane representations

| Label | Rotational symmetry | Permutational symmetry | #Features |
|---|---|---|---|
| Cartesian Components | Augmentations | Augmentations | 12 |
| Distances | Features | Augmentations | 10 |
| Cartesian Components Polynomials | Augmentations | Features | 34 |
| Distances Polynomials | Features | Features | 31 |

The target, the potential energy of the system, is invariant with respect to rotations and permutations of identical atoms[3]. Therefore, there are several viable representations of the methane molecules depending on how these symmetries are addressed:

`Cartesian Components:` The simplest representation is just a collection of all the Cartesian components of all displacement vectors from the carbon atom to all the hydrogen atoms. Since each methane molecule contains 4 hydrogen atoms, the total number of displacement vectors is 4, and the total number of components is 12. When using this representation, we simply concatenate all these components together and feed them to a fully connected MLP or lmKAN whose input dimension is 12. This representation is not invariant with respect to both rotations and permutations; thus, we use the corresponding augmentations during training. We randomly permute hydrogen atoms and rotate each molecule whenever we sample a minibatch from the training subset for each step of stochastic gradient descent.

`Distances:` Another possible representation is a collection of all the interatomic distances between all the atoms. Since the total number of atoms is 5, the number of all the interatomic distances is $5 * 4/2 = 10$. Therefore, the input dimension of fully connected networks applied to this representation is 10. This representation is invariant with respect to rotations but not with respect to permutations. During training, we use only permutational augmentations.

---

[3]Formally, there is an additional symmetry, inversion, but the corresponding group contains only two elements, thus it does not make much sense to treat it separately. We unite it with the group of rotations, and in the following, by rotation we mean proper or improper rotation.

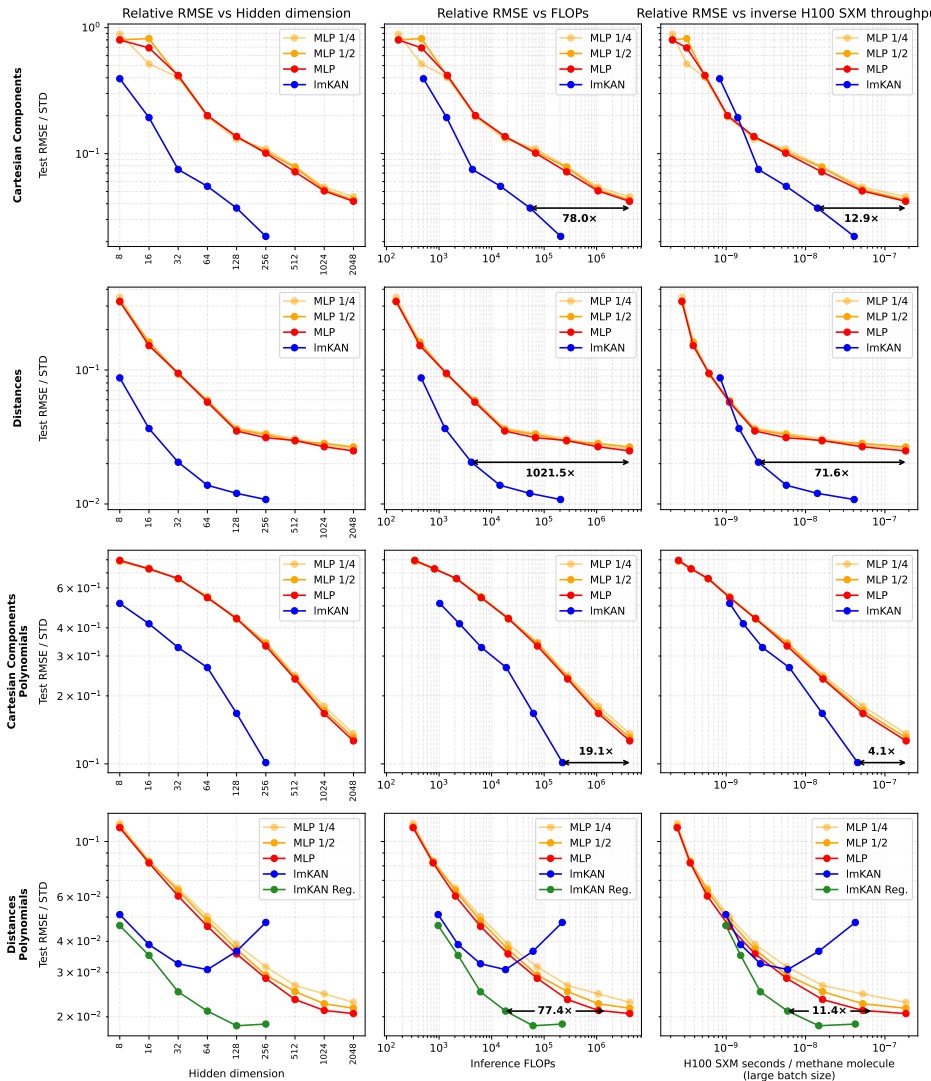

Figure 22: lmKAN vs MLP on the dataset of randomly displaced methane configurations. "lmKAN Reg." curve corresponds to lmKAN fitted with Hessian regularization introduced in Appendix D. On the vertical axis, we plot the relative Root Mean Squared Error, which is given as test RMSE normalized by the standard deviation of the target in the dataset. The "MLP 1/2" and "MLP 1/4" curves correspond to outcomes of fitting procedures with half and a quarter of the training budget, respectively.

**Cartesian Components Polynomials:** We compute power sum symmetric polynomials on top of the Cartesian components of the displacement vectors: $P_{\alpha_x,\alpha_y,\alpha_z} = \sum_{i=1}^{i=4} x_i^{\alpha_x} y_i^{\alpha_y} z_i^{\alpha_z}$ for non-negative integer $\alpha_x + \alpha_y + \alpha_z \leq 4$. The total number of such symmetric polynomials is 34 (excluding trivial $P_{0,0,0}$). This representation is invariant with respect to permutations but not with respect to rotations. Thus, during training we use only rotational augmentations.

**Distances Polynomials:** The final representation is a collection of non-trivial symmetric polynomials on top of the interatomic distances, constructed similarly to Allen et al. (2021). The total number of such polynomials is 31, and their exact formulas are given in Appendix G.4. This representation is invariant with respect to both rotations and permutations. Thus, we do not use any augmentations during training for this representation.

The described representations are summarized in Table 1. We systematically evaluate all four possible combinations of how the rotational and permutational symmetries can be

incorporated into the fitting pipeline. Within the `Distances Polynomials` representation, the methane dataset is tabular in the classical sense — it is a table with about 7.7 million rows and 31 columns. For other representations, the dataset is tabular-like given the available augmentation strategies. We randomly split the data into `7000000`, `300000`, and `432488` train, validation, and test molecules, respectively.

For each representation, we fit the same families of MLP and lmKAN models as in the previous section. The result is given in Fig. 22. For this dataset, we use $G = 28$, the optimal value we found in ablation studies. Similarly to the previous experiment, we demonstrate tight convergence of the baseline MLP models by providing three lines corresponding to full, half, and quarter of the training budget, respectively. Overall, when compared to domain-specific architectures, typically given by GNNs (Zhang et al., 2021) and/or transformers (Pozdnyakov & Ceriotti, 2023), the introduced feedforward fully connected models occupy a non-overlapping part of the Pareto frontier — they are less accurate, but also orders of magnitude faster.

The figure illustrates that lmKANs consistently outperform MLPs across all modalities. Furthermore, the performance improvement is much larger compared to our previous experiment. At the same accuracy level, lmKANs require up to many dozen times (or even more for the `Distances` modality) less inference FLOPs, which results in **more than a 10×** improvement of the inference H100 wall-clock time.

Furthermore, Fig. 22 provides early indications that lmKANs sometimes can be more accurate in the limit of large scale, that is, to have better generalizability. The second row of the figure, corresponding to the `Distances` modality, illustrates that the rate of improvement of MLP models becomes very slow, and it is questionable if this family of models would ever surpass the accuracy achieved by lmKAN models at any scale.

On the other hand, depending on the nature of the data, raw lmKANs, without the Hessian regularization we proposed in Appendix D, can be more prone to overfitting. This happens for the `Distances Polynomials` modality as the last row of Fig. 22 illustrates. This modality incorporates all the symmetries into the representation and does not involve any sort of augmentations. Therefore, it is likely that the generalization problem we outlined in Appendix D takes place for this fitting setup. As the green line of the fourth row of Fig. 22 illustrates, the Hessian regularization is sufficient to overcome the overfitting. Properly regularized lmKANs were found to outperform the MLPs and be Pareto-optimal from the point of view of both inference FLOPs and inference H100 wall-clock time.

### G.4.1 Ablations

The ablation study on the effect of Hessian regularization is given in Fig. 23, and the effect of the number of grid intervals $G$ in Fig. 24.

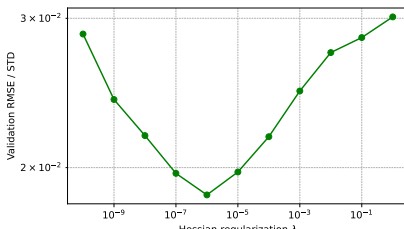

Figure 23: Effect of the strength of Hessian regularization on the validation error when fitting lmKAN with hidden_dim = 256 on the methane dataset using the `Distances Polynomials` representation.

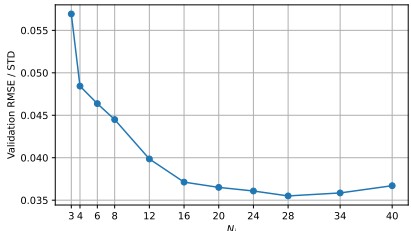

Figure 24: Effect of the number of grid intervals $G$ on the validation error when fitting lmKAN with hidden_dim = 128 on the methane dataset using the Cartesian Components representation.

### G.4.2 THE `Distances Polynomials` REPRESENTATION

It was mentioned that the `Distances Polynomials` representation is given by non-trivial invariant polynomials computed on top of interatomic distances. These polynomials are constant with respect to changing the order of identical hydrogen atoms.

$$P_1 = x_5 + x_6 + x_7 + x_8 + x_9 + x_{10}$$

$$P_2 = x_1 + x_2 + x_3 + x_4$$

$$P_3 = x_5^2 + x_6^2 + x_7^2 + x_8^2 + x_9^2 + x_{10}^2$$

$$P_4 = x_5 x_6 + x_5 x_7 + x_6 x_7 + x_5 x_8 + x_6 x_8 + x_5 x_9 + x_7 x_9 + x_8 x_9 + \\ x_6 x_{10} + x_7 x_{10} + x_8 x_{10} + x_9 x_{10}$$

$$P_5 = x_1 x_5 + x_2 x_5 + x_1 x_6 + x_3 x_6 + x_1 x_7 + x_4 x_7 + x_2 x_8 + x_3 x_8 + \\ x_2 x_9 + x_4 x_9 + x_3 x_{10} + x_4 x_{10}$$

$$P_6 = x_1^2 + x_2^2 + x_3^2 + x_4^2$$

$$P_7 = x_5^3 + x_6^3 + x_7^3 + x_8^3 + x_9^3 + x_{10}^3$$

$$P_8 = x_5^2 x_6 + x_5 x_6^2 + x_5^2 x_7 + x_6^2 x_7 + x_5 x_7^2 + x_6 x_7^2 + x_5^2 x_8 + x_6^2 x_8 + \\ x_5 x_8^2 + x_6 x_8^2 + x_5^2 x_9 + x_7^2 x_9 + x_8^2 x_9 + x_5 x_9^2 + x_7 x_9^2 + x_8 x_9^2 + \\ x_6^2 x_{10} + x_7^2 x_{10} + x_8^2 x_{10} + x_9^2 x_{10} + x_6 x_{10}^2 + x_7 x_{10}^2 + x_8 x_{10}^2 + x_9 x_{10}^2$$

$$P_9 = x_1 x_5^2 + x_2 x_5^2 + x_1 x_6^2 + x_3 x_6^2 + x_1 x_7^2 + x_4 x_7^2 + x_2 x_8^2 + x_3 x_8^2 + \\ x_2 x_9^2 + x_4 x_9^2 + x_3 x_{10}^2 + x_4 x_{10}^2$$

$$P_{10} = x_5 x_6 x_8 + x_5 x_7 x_9 + x_6 x_7 x_{10} + x_8 x_9 x_{10}$$

$$P_{11} = x_1 x_5 x_6 + x_1 x_5 x_7 + x_1 x_6 x_7 + x_2 x_5 x_8 + x_3 x_6 x_8 + x_2 x_5 x_9 + x_4 x_7 x_9 + x_2 x_8 x_9 + \\ x_3 x_6 x_{10} + x_4 x_7 x_{10} + x_3 x_8 x_{10} + x_4 x_9 x_{10}$$

$$P_{12} = x_1^2 x_5 + x_2^2 x_5 + x_1^2 x_6 + x_3^2 x_6 + x_1^2 x_7 + x_4^2 x_7 + x_2^2 x_8 + x_3^2 x_8 + \\ x_2^2 x_9 + x_4^2 x_9 + x_3^2 x_{10} + x_4^2 x_{10}$$

$$P_{13} = x_1 x_2 x_5 + x_1 x_3 x_6 + x_1 x_4 x_7 + x_2 x_3 x_8 + x_2 x_4 x_9 + x_3 x_4 x_{10}$$

$$P_{14} = x_1^3 + x_2^3 + x_3^3 + x_4^3$$

$$P_{15} = x_5^4 + x_6^4 + x_7^4 + x_8^4 + x_9^4 + x_{10}^4$$

$$P_{16} = x_5^3 x_6 + x_5 x_6^3 + x_5^3 x_7 + x_6^3 x_7 + x_5 x_7^3 + x_6 x_7^3 + x_5^3 x_8 + x_6^3 x_8 + \\ x_5 x_8^3 + x_6 x_8^3 + x_5^3 x_9 + x_7^3 x_9 + x_8^3 x_9 + x_5 x_9^3 + x_7 x_9^3 + x_8 x_9^3 + \\ x_6^3 x_{10} + x_7^3 x_{10} + x_8^3 x_{10} + x_9^3 x_{10} + x_6 x_{10}^3 + x_7 x_{10}^3 + x_8 x_{10}^3 + x_9 x_{10}^3$$

$$P_{17} = x_1 x_5^3 + x_2 x_5^3 + x_1 x_6^3 + x_3 x_6^3 + x_1 x_7^3 + x_4 x_7^3 + x_2 x_8^3 + x_3 x_8^3 + \\ x_2 x_9^3 + x_4 x_9^3 + x_3 x_{10}^3 + x_4 x_{10}^3$$

$$P_{18} = x_1 x_5^2 x_6 + x_1 x_5 x_6^2 + x_1 x_5^2 x_7 + x_1 x_6^2 x_7 + x_1 x_5 x_7^2 + x_1 x_6 x_7^2 + x_2 x_5^2 x_8 + x_3 x_6^2 x_8 + \\ x_2 x_5 x_8^2 + x_3 x_6 x_8^2 + x_2 x_5^2 x_9 + x_4 x_7^2 x_9 + x_2 x_8^2 x_9 + x_2 x_5 x_9^2 + x_4 x_7 x_9^2 + x_2 x_8 x_9^2 + \\ x_3 x_6^2 x_{10} + x_4 x_7^2 x_{10} + x_3 x_8^2 x_{10} + x_4 x_9^2 x_{10} + x_3 x_6 x_{10}^2 + x_4 x_7 x_{10}^2 + x_3 x_8 x_{10}^2 + x_4 x_9 x_{10}^2$$

$$P_{19} = x_2 x_5^2 x_6 + x_3 x_5 x_6^2 + x_2 x_5^2 x_7 + x_3 x_6^2 x_7 + x_4 x_5 x_7^2 + x_4 x_6 x_7^2 + x_1 x_5^2 x_8 + x_1 x_6^2 x_8 + \\ x_3 x_5 x_8^2 + x_2 x_6 x_8^2 + x_1 x_5^2 x_9 + x_1 x_7^2 x_9 + x_3 x_8^2 x_9 + x_4 x_5 x_9^2 + x_2 x_7 x_9^2 + x_4 x_8 x_9^2 + \\ x_1 x_6^2 x_{10} + x_1 x_7^2 x_{10} + x_2 x_8^2 x_{10} + x_2 x_9^2 x_{10} + x_4 x_6 x_{10}^2 + x_3 x_7 x_{10}^2 + x_4 x_8 x_{10}^2 + x_3 x_9 x_{10}^2$$

$$P_{20} = x_1^2 x_5^2 + x_2^2 x_5^2 + x_1^2 x_6^2 + x_3^2 x_6^2 + x_1^2 x_7^2 + x_4^2 x_7^2 + x_2^2 x_8^2 + x_3^2 x_8^2 + \\ x_2^2 x_9^2 + x_4^2 x_9^2 + x_3^2 x_{10}^2 + x_4^2 x_{10}^2$$

$$(12)$$

The exact form of these polynomials is given in Eq. 12 and Eq. 13, where $x_1$, $x_2$, ... $x_{10}$ correspond to interatomic distances $CH_1$, $CH_2$, $CH_3$, $CH_4$, $H_1 H_2$, $H_1 H_3$, $H_1 H_4$, $H_2 H_3$,

$H_2H_4$, and $H_3H_4$ respectively. The presented polynomials form invariant generators (Derksen & Kemper, 2015) of the group corresponding to arbitrary permutations of the hydrogen atoms. Therefore, the `Distances Polynomials` representation preserves all the information about the initial $CH_4$ molecule.

$$P_{21} = x_1x_2x_5^2 + x_1x_3x_6^2 + x_1x_4x_7^2 + x_2x_3x_8^2 + x_2x_4x_9^2 + x_3x_4x_{10}^2$$

$$P_{22} = x_1^2x_5x_6 + x_1^2x_5x_7 + x_1^2x_6x_7 + x_2^2x_5x_8 + x_3^2x_6x_8 + x_2^2x_5x_9 + x_4^2x_7x_9 + x_2^2x_8x_9 +$$
$$x_3^2x_6x_{10} + x_4^2x_7x_{10} + x_3^2x_8x_{10} + x_4^2x_9x_{10}$$

$$P_{23} = x_1^3x_5 + x_2^3x_5 + x_1^3x_6 + x_3^3x_6 + x_1^3x_7 + x_4^3x_7 + x_2^3x_8 + x_3^3x_8 +$$
$$x_2^3x_9 + x_4^3x_9 + x_3^3x_{10} + x_4^3x_{10}$$

$$P_{24} = x_1^4 + x_2^4 + x_3^4 + x_4^4$$

$$P_{25} = x_5^5 + x_6^5 + x_7^5 + x_8^5 + x_9^5 + x_{10}^5$$

$$P_{26} = x_1x_5^4 + x_2x_5^4 + x_1x_6^4 + x_3x_6^4 + x_1x_7^4 + x_4x_7^4 + x_2x_8^4 + x_3x_8^4 +$$
$$x_2x_9^4 + x_4x_9^4 + x_3x_{10}^4 + x_4x_{10}^4$$

$$P_{27} = x_1x_5^3x_6 + x_1x_5x_6^3 + x_1x_5^3x_7 + x_1x_6^3x_7 + x_1x_5x_7^3 + x_1x_6x_7^3 + x_2x_5^3x_8 + x_3x_6^3x_8 +$$
$$x_2x_5x_8^3 + x_3x_6x_8^3 + x_2x_5^3x_9 + x_4x_7^3x_9 + x_2x_8^3x_9 + x_2x_5x_9^3 + x_4x_7x_9^3 + x_2x_8x_9^3 +$$
$$x_3x_6^3x_{10} + x_4x_7^3x_{10} + x_3x_8^3x_{10} + x_4x_9^3x_{10} + x_3x_6x_{10}^3 + x_4x_7x_{10}^3 + x_3x_8x_{10}^3 + x_4x_9x_{10}^3$$

$$P_{28} = x_1^2x_5^3 + x_2^2x_5^3 + x_1^2x_6^3 + x_3^2x_6^3 + x_1^2x_7^3 + x_4^2x_7^3 + x_2^2x_8^3 + x_3^2x_8^3 +$$
$$x_2^2x_9^3 + x_4^2x_9^3 + x_3^2x_{10}^3 + x_4^2x_{10}^3$$

$$P_{29} = x_1x_2x_5^3 + x_1x_3x_6^3 + x_1x_4x_7^3 + x_2x_3x_8^3 + x_2x_4x_9^3 + x_3x_4x_{10}^3$$

$$P_{30} = x_1^3x_5^2 + x_2^3x_5^2 + x_1^3x_6^2 + x_3^3x_6^2 + x_1^3x_7^2 + x_4^3x_7^2 + x_2^3x_8^2 + x_3^3x_8^2 +$$
$$x_2^3x_9^2 + x_4^3x_9^2 + x_3^3x_{10}^2 + x_4^3x_{10}^2$$

$$P_{31} = x_1^3x_2x_5 + x_1x_2^3x_5 + x_1^3x_3x_6 + x_1x_3^3x_6 + x_1^3x_4x_7 + x_1x_4^3x_7 + x_2^3x_3x_8 + x_2x_3^3x_8 +$$
$$x_2^3x_4x_9 + x_2x_4^3x_9 + x_3^3x_4x_{10} + x_3x_4^3x_{10}$$

$$\tag{13}$$

### G.5 lmKAN-based Convolutional Neural Networks

For CIFAR-10, our backbone architecture consists of five $2 \times 2$ convolutions, each with stride 2, and two fully connected layers at the end. Since the resolution of CIFAR-10 images is $32 \times 32$, where $32 = 2^5$, five $2 \times 2$ convolutions with stride 2 transform the spatial dimensions of an image exactly to $1 \times 1$. All the layers use the same width (= number of filters in case of convolutions, and hidden dimension in case of fully connected layers), which we vary for both families of the models. In other aspects, the models are similar to those we employed in previous sections - we use batch normalizations with affine transforms for MLP-CNNs, and without for lmKAN-CNNs; MLP-CNNs use ReLU activations, while lmKAN-CNNs do not require additional activation layers.

The dataset comes with pre-defined full training and test subsets. We split the full training subset into training and validation parts in a 90%/10% ratio. Our augmentation pipeline consists of established techniques, such as RandAugment Cubuk et al. (2020), MixUp Zhang et al. (2017), CutMix Yun et al. (2019), and a few others.

**CIFAR-10 CNN backbone**

```
# Only convolutional and fully
    connected layers are shown

# [32, 32, 3] → [16, 16, width]
Conv2D(3 → width, kernel_size = 2,
    stride = 2)

# [16, 16, width] → [8, 8, width]
Conv2D(width → width, kernel_size =
    2, stride = 2)

# [8, 8, width] → [4, 4, width]
Conv2D(width → width, kernel_size =
    2, stride = 2)

# [4, 4, width] → [2, 2, width]
Conv2D(width → width, kernel_size =
    2, stride = 2)

# [2, 2, width] → [1, 1, width]
Conv2D(width → width, kernel_size =
    2, stride = 2)

FullyConnected(width → width)
FullyConnected(width → 10)
```

**ImageNet CNN backbone**

```
# Only convolutional and fully
    connected layers are shown

# [81, 81, 3] → [27, 27, base_width]
Conv2D(3 → base_width, kernel_size =
    3, stride = 3)

# [27, 27, base_width] → [9, 9, 3*
    base_width]
Conv2D(base_width → 3*base_width,
    kernel_size = 3, stride = 3)

# [9, 9, 3*base_width] → [3, 3, 9*
    base_width]
Conv2D(3*base_width → 9*base_width,
    kernel_size = 3, stride = 3)

# [3, 3, 9*base_width] → [1, 1, 27*
    base_width]
Conv2D(9*base_width → 27*base_width,
    kernel_size = 3, stride = 3)

FullyConnected(27*base_width → 27*
    base_width)
FullyConnected(27*base_width → 1000)
```

Figure 25: CIFAR-10 and ImageNet CNN backbones. MLP-based CNNs additionally have ReLU activations and batch normalizations with enabled affine transforms. lmKAN-based CNNs do not require additional activations and use batch normalizations without affine transforms as suggested by our sigma grids described in Sec. 3.1.

We use the following pool of augmentations for the CIFAR-10 dataset:

**CIFAR-10 augmentation pipeline**

```
MEAN = (0.4914, 0.4822, 0.4465)
STD = (0.2470, 0.2435, 0.2616)

nn.Sequential(
    T.RandomCrop(32, padding=4),
    T.RandomHorizontalFlip(),
    T.ColorJitter(0.3, 0.3, 0.3, 0.05),
    T.RandAugment(2, 7),
    T.RandomErasing(p=0.25, scale=(0.05, 0.2), ratio=(0.3, 3.3)),
    T.Normalize(MEAN, STD),
)
```

On top of these, we use MixUp (Zhang et al., 2017) ($\alpha = 0.2$) and CutMix (Yun et al., 2019) ($\beta = 1.0$) augmentations, both with 50% probability.

When fitting the families of convolutional neural networks described in the main text, we use the above pool of augmentations consistently for MLP-based and lmKAN-based CNNs.

For ImageNet, the standard data preparation pipeline introduced by AlexNet(Krizhevsky et al., 2012) involves first resizing an image to 256 pixels along the smallest dimension, then performing a random crop of $224 \times 224$ pixels during training, and a center crop of $224 \times 224$ pixels during validation.

We mimic this procedure by first resizing the image to $81 \cdot 256/224 \approx 93$ pixels across the smallest dimension, and then performing random or center crops of $81 \times 81$ pixels.

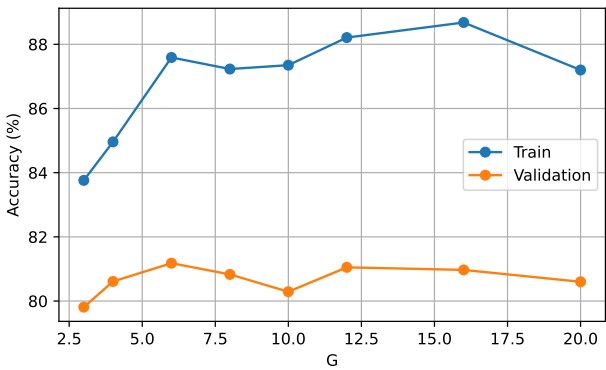

Figure 26: Accuracy of the lmKAN-based CNNs on the CIFAR-10 dataset depending on the grid resolution G.

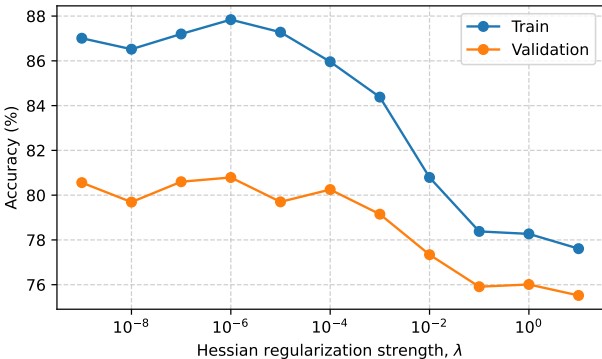

Figure 27: Accuracy of the lmKAN-based CNNs on the CIFAR-10 dataset depending on the strength of Hessian regularization.

Next, we use the following augmentation pipeline:

**ImageNet augmentation pipeline**

```python
nn.Sequential(
    T.RandomHorizontalFlip(),
    T.ColorJitter(brightness=0.4, contrast=0.4, saturation=0.4, hue=0.1),
    T.RandAugment(),
    T.RandomErasing(p=0.25, scale=(0.02, 0.33), ratio=(0.3, 3.3), value=0),
    T.Normalize(mean=[0.485, 0.456, 0.406], std=[0.229, 0.224, 0.225]),
)
```

On top of these, we use MixUp and CutMix.

## G.6 COMPARISON WITH FASTKAN

As was already mentioned in the main text, we use the training script for the CIFAR-10 dataset available in the FastKAN GitHub repository (Li, 2024a) as the basis for the comparison of lmKAN and FastKAN. However, we provide several modifications to the pipeline.

The original script implements the fitting procedure of a fully connected FastKAN model on the CIFAR-10 dataset without augmentations. The model has only one hidden layer

with 256 neurons. Without augmentations, it overfits the data quickly. Thus, the very short fitting procedure in the original script is sufficient.

We extend the script by the same augmentation pipeline as we used in Sec. 4.3 for the CIFAR-10 dataset. We observed that because of augmentations, one has to fit the model for a longer time, so the training budget was substantially increased. The data was split properly into train, validation, and test subsets, while the original script employed only a train-validation split. We use the cosine (without restarts) learning rate scheduler (Loshchilov & Hutter, 2016) instead of the exponential decay one in the original script. Finally, the normalization was performed with real values of the mean and standard deviation for the CIFAR-10 dataset, instead of the dummy 0.5 values of the original script.

These modifications significantly improve the performance of FastKAN models ($\approx 54 - 55\%$ validation accuracy in the original script), see Fig. 9. Furthermore, even the MLP baseline of the same shape yields better accuracy compared to the performance of the FastKAN model in the original script.

