# OpenReview forum: "Lookup multivariate Kolmogorov-Arnold Networks"
_ICLR.cc/2026/Conference — ICLR 2026 Poster_

### Official Review · Reviewer_jcL7 · 2025-10-17

**Soundness:** 3
**Presentation:** 3
**Contribution:** 3
**Rating:** 6
**Confidence:** 4

**Summary:**

This paper proposes to increase KANs' inference efficiency via lookup tables. Their construction expresses a general high-dimensional mapping through trainable low-dimensional multivariate functions. Although these functions may carry a large number of trainable parameters, they sparsely activate during inference time; as a result, it can be implemented as a sparse lookup table. They show the effectiveness of the proposed method on a wide range of datasets, outperforming MLPs and FastKANs.

**Strengths:**

* The paper is well written and the presentation is good
* The proposed sparse lookup table method is simple yet effective
* Experiments are extensive and convincing

**Weaknesses:**

* The application range of the method should be discussed, i.e., when do we expect this method to be less of an advantage?
* Although the proposed method can increase inference efficiency, it does not necessarily help training when a larger than 1 batch is being used.

**Questions:**

* The default setup is $d=2$, how about $d=1$ or $d=3$ or larger? What's the philosophy for choosing $d=2$?
* How about training efficiency? When the batch is larger than 1, different samples can activate different basis functions.
* Section 4.3, it should be "IMKAN", not "LMKAN"

---

> ### Author Response · Authors · 2025-11-23
>
> We thank the reviewer for the detailed feedback, which we used to improve our paper.
>
> Below, we address the reviewer's concerns and highlight the relevant sections of the revised manuscript.
>
> ---
>
> > **Weakness 1**
>
> This is a critical question. Our extensive experiments across diverse setups were designed specifically to answer it. We found that lmKANs achieve the most significant performance gains on the methane dataset, followed by the random function approximation task (modeled as distillation from a large, random-weight teacher MLP). While the relative improvement was smallest for CNN models, it remains substantial: we observed a 1.6x to 2.1x reduction in inference FLOPs for the same accuracy on the CIFAR-10 and ImageNet datasets.
>
> > **Weakness 2, Question 2**
>
> The computational cost of both the forward and backward passes of lmKANs doesn't depend on grid resolution G for any batch size. When the batch is larger than 1, different samples can indeed activate different basis functions, but the effect of the activation of the corresponding basis function is contained within the given sample and doesn't affect the other samples within the batch. Overall, the computations required for both forward and backward passes are independent of each other for each sample in the batch. For the forward pass, more details are available in Algorithm 3 in Appendix C. The computational pipeline for the backward pass mirrors that for the forward one.
>
> > **Question 1**
>
> In the revised manuscript, we have implemented additional CUDA kernels for the 1D case and empirically compared the performance of 1D and 2D lmKANs (see Figures 4 and 9 and the associated discussion).
>
> We have also moved the section "Perspective on dimensions and spline orders" to the main text to better motivate this design choice. The inference FLOPs of a $d$-dimensional lmKAN model scale as $2^d/d$ relative to a linear layer of the same shape. This results in an identical inference cost multiplier ($2\times$) for both 1D and 2D lmKANs, whereas the cost begins to grow for higher dimensions. Therefore, it was reasonable to prioritize optimized CUDA kernels for $d=1$ and $d=2$ over higher dimensions.
>
> > **Question 3**
>
> We thank the reviewer for a good catch. We fixed the typo in the revised version of the manuscript.
>
> ---
>
> We thank the reviewer again for their detailed feedback and look forward to their reply.

---

### Official Review · Reviewer_63pm · 2025-10-30

**Soundness:** 1
**Presentation:** 2
**Contribution:** 2
**Rating:** 2
**Confidence:** 5

**Summary:**

The paper introduces lookup multivariate Kolmogorov-Arnold Networks (lmKANs). The central idea is to replace the 1D univariate activation functions used in standard KANs with 2D multivariate functions. The key component of this architecture is the implementation of these 2D functions as $\mathcal{O}(1)$ spline lookup tables. The authors posit that this approach "decouples" the model's parameter capacity which could be increased by grid point from its inference cost. This, in theory, allows lmKANs to possess a significantly larger parameter count than an MLP with only a $\sim 2\times$ increase in theoretical FLOPs. The paper presents experiments on general function approximation, the methane molecule dataset, and image classification to demonstrate that lmKANs achieve a superior trade-off between accuracy and inference cost compared to MLPs.

**Strengths:**

- The idea of using low-dimensional multivariate functions (specifically 2D) implemented as $\mathcal{O}(1)$ spline lookup tables is novel.
- The authors have clearly put significant effort into the implementation, including custom CUDA kernels and a sophisticated multi-stage training pipeline with Hessian regularization.

**Weaknesses:**

- The paper's primary thesis is that lmKANs are superior because parameters can be added "for free" in terms of FLOPs by increasing the grid resolution $G$. This is the claimed advantage over MLPs, where adding parameters requires increasing width $N$ at an $\mathcal{O}(N^2)$ FLOPs cost. However, the main experimental comparisons (Figs 4, 6, 7) do not test this thesis. These experiments fix $G$ and vary $N$, just like an MLP. These results show, at best, that lmKAN's $\mathcal{O}(N^2)$ scaling factor is better than an MLP's, but they provide zero evidence for the central claim about the benefit of scaling $G$.
- The most important experiment is missing from the paper. To validate the core claim, the authors must provide an analysis where network width $N$ is fixed, while grid resolution $G$ is varied (e.g., $G \in \{4, 8, 16, 32\}$). This experiment must plot two curves on the same y-axis: (1) model performance (e.g., MSE) and (2) inference cost (FLOPs and/or wall-clock time). The paper's hypothesis predicts that performance will improve while inference cost remains flat. Without this plot, the main premise is entirely unsubstantiated.
- The only analysis of $G$ is relegated to the appendix (e.g., Fig 19, 22, 24). These plots are incomplete, showing only "Performance vs. $G$". They critically omit the corresponding "Inference Cost vs. $G$" data, which is essential to validating the trade-off. Furthermore, these plots (e.g., Fig 19) show a U-shaped curve where performance worsens after $G=12$, which contradicts the paper's narrative that multivariate functions can "accommodate" large parameters without issue. This is not adequately discussed.
- The paper motivates the move from 1D to 2D functions by claiming 1D functions with dense grids are "unstable". To support this, it compares its 2D B-spline lmKAN to FastKAN (Fig 8), which uses 1D Gaussian RBFs. This is an invalid. The degradation in FastKAN could be a property of RBFs, not a fundamental flaw of 1D spline functions. The correct baseline would be a 1D B-spline KAN using the same $\mathcal{O}(1)$ lookup technology. This baseline is missing.

**Questions:**

- Can the authors please provide the critical experiment described in Weakness #2? Specifically, for the "General Function Approximation" task (from Fig 4), please provide a plot with a fixed hidden dimension (e.g., $N=256$) while sweeping $G$ (e.g., $G \in \{4, 8, 16, 32\}$). This plot must show (a) Test MSE vs. $G$ and (b) Inference FLOPs vs. $G$. This is essential to validate the paper's core premise.
- Why did the authors choose to compare against the RBF-based FastKAN to justify the 1D vs. 2D claim, rather than the much more appropriate baseline of a 1D B-spline KAN using the same lookup table method? Without this direct comparison, the claim that 2D functions are inherently more stable is unsupported.
- The appendix figures (e.g., Fig 19) show that performance degrades as $G$ becomes very large. This seems to contradict the motivation that multivariate functions can "accommodate" a large number of parameters without issue. Does this imply that the "free parameter" benefit is limited in practice? How do the authors reconcile this with their primary motivation?

---

> ### Author Response · Authors · 2025-11-23
>
> We thank the reviewer for the detailed feedback and concrete suggestions for improving our paper.
>
> Below, we discuss how the revised version of our manuscript addresses the concerns raised by the reviewer.
>
> ---
>
> > **Weakness 1, 2; Question 1.**
>
> We have included the requested analysis in **Figure 4** of the revised manuscript. This figure simultaneously plots 1) Mean Squared Error (MSE), 2) inference FLOPs, and 3) inference H100 wall-clock time as a function of the number of trainable parameters per each inner function or the grid resolution $G$.
>
> The results demonstrate that neither inference FLOPs nor H100 wall-clock time depends on the grid resolution $G$. This substantiates our claim that lmKAN allows for the addition of trainable parameters without increasing computational cost. As the left panel illustrates, these additional parameters reduce MSE: 2D lmKAN significantly outperforms an MLP with the same hidden dimension and achieves performance comparable to an MLP with $4\times$ the hidden dimension (which is, thus, approximately $4^2=16$ times larger).
>
> We have also included a vertical version of this plot in **Appendix G.3**, where the x-axis ($G$, or parameters per inner function) is shared across all subplots, as requested. We placed it in the appendix because of the main text's one-column format.
>
> Figure 4 additionally includes the performance of 1D lmKANs to address the reviewer's other concerns.
>
> > **Weakness 3; Question 3.**
>
> To address this concern, we implemented CUDA kernels for 1D lmKANs and analyzed their performance as a function of grid resolution $G$, parallel to our 2D analysis (also shown in Fig. 4). While both 1D and 2D lmKANs exhibit U-shaped performance curves, there are substantial differences:
> 1. 2D lmKAN performance peaks at $(G+1)^2=169$ parameters per inner function (for $G=12$), whereas 1D lmKAN performance starts to degrade after just $(G+1)=17$ parameters (for $G=16$).
> 2. The peak performance of 2D lmKANs is noticeably better than that of 1D lmKANs.
>
> These findings demonstrate that multivariate inner functions can meaningfully leverage a (much) larger number of parameters and are more accurate overall compared to the one-dimensional functions used in the standard formulation of KANs.
>
> This conclusion is further supported by our updated comparison with FastKAN, which now additionally includes 1D lmKANs.
>
> > These results show, at best, that lmKAN's O(N^2) scaling factor is better than an MLP's. (and Question  3)
>
> Linear layers (high-dimensional mappings) are the core building blocks of most modern deep learning architectures. Consequently, developing a construction with a substantially better prefactor is a highly impactful contribution and remains the central claim of our paper. We state (e.g., in the abstract) that lmKANs "deliver a **substantially** better trade-off between capacity and inference cost," rather than an **unlimitedly** better trade-off compared to MLPs.
>
> > **Weakness 4; Question 2**
>
> We initially compared 2D lmKANs with FastKAN, a state-of-the-art KAN variant, to demonstrate competitive performance. In the revised manuscript, we have explicitly added 1D lmKANs to this comparison as requested.
>
> **Figure 9** of the revised manuscript compares all three models: 2D lmKANs, 1D lmKANs, and FastKANs (1D). Overall, 1D lmKANs exhibit similar behavior to FastKANs, with performance degrading at excessively high grid resolutions $G$. This degradation is even more pronounced for 1D lmKANs than for FastKANs (which parameterize 1D functions using Gaussian RBFs). This updated comparison underscores that 2D lmKANs are more accurate and easier to train than both 1D lmKANs and FastKANs, thus supporting that multivariate inner functions as building blocks are more stable in training and lead to better accuracy compared to univariate functions used in the standard KAN formulation.
>
> ---
>
> We thank the reviewer again for their detailed feedback and constructive suggestions. We look forward to your reply.

---

### Official Review · Reviewer_HStq · 2025-10-31

**Soundness:** 4
**Presentation:** 3
**Contribution:** 3
**Rating:** 6
**Confidence:** 4

**Summary:**

This paper proposes lookup multivariate Kolmogorov-Arnold Networks (lmKANs), a novel architecture that replaces high-dimensional linear mappings with trainable low-dimensional multivariate functions (primarily d=2) implemented via spline lookup tables.

lmKANs split high-dimensional inputs into low-dimensional subgroups (e.g., 32D input → 16×2D subgroups), each processed by a multivariate spline function.
Spline lookup tables leverage the local support of B-splines to achieve O(1) inference per subgroup. This design reduces inference FLOPs by up to 6× for general function approximation.

Custom CUDA kernels further accelerate computation, achieving up to 88× faster inference on H100 GPUs. Across tasks, lmKANs consistently outperform MLPs and FastKANs in both accuracy and efficiency.

**Strengths:**

The idea is original and overall design logics is very well organized. By grouping high-dimensional inputs into low-dimensional subgroups, lmKANs avoid the exponential complexity of high-dimensional splines. The look-up table take advantage of this The Sigmoid-like σ-grid very elegant way to adapts to unbound activations into bounded region.

Experiments are solid: across four task types, the paper tests different model scales, training budgets, and data formats to show lmKANs’ robustness.
Engineering details, including CUDA implementation, are clear too.

Also, the paper does a good job of explaining a relatively complex idea in a clear and accessible way.

**Weaknesses:**

Limited high-dimensional multivariate validation
The paper focuses exclusively on 2D lmKANs. It would provide values to validate for other d=4/8. Even on very small experiments this would help.

CNN architecture and  implementation
The experiments on CIFAR uses “For CIFAR-10, our backbone architecture consists of five 2 × 2 convolutions, each with stride 2” and for ImageNet “For ImageNet, we downsample the images to a resolution of 81 × 81 pixels. Our backbone consists of four convolutional layers with the 3 × 3 kernel size and stride 3 and two fully connected layers” These are not typical architectures for these datasets. Does this imply lmKANs cannot integrate with standard models like ResNet or AlexNet?

Additionally, Section 4.3 mentions that lmKAN-CNNs “cast convolutions to fully connected layers via memory manipulations.” This seems like a temporary workaround rather than a native multivariate spline convolution.

**Questions:**

Appendix D mentions that second-order B-splines match ReLU’s smoothness. Did the authors try higher-order splines?

Also, Hessian regularization is only mentioned in the appendix. Was it applied to all experiments in the main paper? If so, how does it affect the results?

Did the author tried other options for mapping unbounded activation to bounded regions? For example, Erf-based function, such that we may assume the activation follows a normal distribution.

---

> ### Author Response · Authors · 2025-11-23
>
> We thank the reviewer for their detailed feedback, which has helped us to improve the paper. Below, we address the reviewer's concerns and highlight relevant revisions in the manuscript.
>
> **Weakness 1**
>
> We have developed additional CUDA kernels for the one-dimensional case and conducted an empirical performance comparison between 1D and 2D lmKANs, which is now detailed in the revised manuscript (see Figures 4 and 9, along with the corresponding discussions).
>
> We also moved the "Perspective on dimensions and spline orders" section into the main text to clarify our design choices. We highlight that the inference FLOPs for a $d$-dimensional lmKAN scale as $2^d/d$ relative to a linear layer of the same shape. Consequently, the inference overhead is identical ($2\times$) for both $d=1$ and $d=2$, but increases for higher dimensions. This identifies $d=1$ and $d=2$ as the most practical targets for initial CUDA optimization. It also raises the question of whether the potential benefits of higher-dimensional inner functions justify both the increased computational overhead and the significantly greater implementation complexity compared to the 1D and 2D cases.
>
> **Weaknesses 2 and 3**
>
> Developing efficient native CUDA kernels for lmKAN-based convolutions is a significant engineering undertaking. To illustrate the complexity of high-performance CUDA programming, one might consider the extensive engineering required "just" to optimize dense matrix multiplications (e.g., as discussed in [this blog](https://www.aleksagordic.com/blog/matmul)) (not affiliated). Therefore, our strategy was to first validate that lmKANs perform effectively within CNN architectures before investing in the development of specialized kernels for lmKAN-based convolutions.
>
> In the interim, a standard workaround for **any** convolution type is to cast it to a fully connected layer using memory manipulations. This approach makes it possible to apply our method also to architectures such as ResNet and AlexNet. The required memory manipulations, however, are the simplest for the specific type of convolutions we used, which justified our choice so far.
>
> **Question 1**
>
> In the "Perspective on dimensions and spline orders" section of the revised manuscript, we discuss why splines of order 2 (piecewise linear functions) are the optimal choice. The computational cost of a $d$-dimensional lmKAN built on splines of order $k$ scales as $k^d/d$ relative to a linear layer—meaning the cost increases dramatically with spline order.
>
> It is well established that splines of any order can approximate any continuous function arbitrarily closely in the limit of grid resolution $G$, with spline order primarily affecting the convergence rate. Since the computational cost of lmKANs is independent of $G$, the most efficient design choice is to use the minimal spline order that still allows gradients to flow through the model.
>
> **Question 2**
>
> For all experiments reported in the main paper, we used Hessian regularization exclusively for preconditioning during the short stability phase at the beginning of training (see Appendix F).
>
> The methane experiment in the main text utilizes Cartesian components of displacement vectors to represent the molecule's geometry. However, other representations are possible; Appendix G.4 details experiments using these alternatives. As shown in Figure 22, unregularized lmKANs overfit quickly when using the "Distance Polynomials" representation. In this specific case, Hessian regularization was crucial, enabling us to solve the problem and achieve generalization performance comparable to MLPs.
>
> **Question 3**
>
> Our choice of sigma functions was driven primarily by computational efficiency; our current construction is significantly less expensive to compute than the error function (`erf`).
>
> While other choices, such as `erf`, might yield better accuracy when the grid resolution $G$ is low, we expect these differences to diminish when $G$ is sufficiently large. Given that increasing $G$ does not incur additional computational cost in our framework, we prioritized the most computationally efficient sigma function variant.
>
> ---
>
> We thank the reviewer again for their constructive comments and look forward to their reply.

---

### Official Review · Reviewer_UnT9 · 2025-10-31

**Soundness:** 2
**Presentation:** 2
**Contribution:** 3
**Rating:** 6
**Confidence:** 3

**Summary:**

This paper introduces lmKANs, a lookup multivariate KAN that replaces the high dimensionality of the univariate functions employed in KANs with multivariate low-dimensional functions, where the inner functions are spline lookup tables. The paper delivers custom CUDA kernels for accelerated inference on lmKANs, conferring 88x faster inference than standard KANs. The paper then compares this implementation with standard MLPs across a range of tasks and demonstrates that lmKANs outperform in terms of accuracy and performance.

**Strengths:**

The paper present an important contribution to KAN development, making KANs easier to train and more scalable. The method presented is novel and is a significant improvement to address hurdles when training KANs.

The paper does a good job benchmarking lmKANs against MLPs. I particularly appreciate the MLP 1/2 traces, giving a sense of the compute benefits afforded by lmKANs.

Overall, the approach is interesting and novel, and provides a way to increase the expressivity of KAN training.

**Weaknesses:**

In the main paper, the authors focus far more on the computational benefit afforded by the accelerations than the theoretical foundation of how this model. However, the supplementary information dives into the theory well and motivates lmKANs, all of which is missing from the main manuscript. This unfortunately detracts from the impact of the main paper. I also struggled a bit to connect the dots between the motivation of the paper and the results presented. Adding dozens or hundreds of more parameters should significantly increase expressivity, but for some plots (e.g. Figure 4) the difference in MSE is on the order of 1e-2, which is surprising given the motivation.

The paper also claims that e.g. Moradzadeh et al do not compare against MLPs, which is untrue, taking a look at the manuscript. This oversight detracts from the novelty of the proposed method, given that other similar approaches to also result in KANs that match performance or outperform MLPs.

While the idea of using a lookup table to convert internal functions to O(1) complexity is interesting, the authors don't explore the added burden of HPO to find the optimal grid size to match or exceed MLP performance.

**Questions:**

Do you have statistics around training time when training lmKANs to convergence versus an equivalent MLP? Figure 18 starts to get at this result with the MLP 1/2 trace, but I would love full training stats.

How sensitive/brittle is training as a function of HPO over G? Is it possible to provide some sensitivity analysis of lmKAN performance to G? I'm trying to understand how much increasing the parameter capacity of KAN's improves expressivity.

---

> ### Author Response · Authors · 2025-11-23
>
> We thank the reviewer for the detailed feedback, which we used to improve our paper.
>
> Below, we address the reviewer's concerns and highlight the relevant sections of the revised manuscript.
>
> ---
>
> > **Weakness 1, not enough theoretical details in the main text**
>
> We have put part of the theoretical analysis in the main text of our paper. Specifically, please find the section "Perspective on dimensions and spline orders", which, also based on feedback from other reviewers, provides the most valuable part of the theoretical discussions.
>
> > **Weakness 1, low absolute difference in MSE**
>
>
> The absolute value of MSE is not an intrinsic indicator of model quality, as it depends entirely on the difficulty of the task and the scale of the target variable. For example, in a regression task where the ground truth has a standard deviation of $10^{-5}$, the difference in MSE between a constant prediction model and a perfect model would be upper-bounded by $\sim10^{-10}$. Small absolute values do not imply that the performance gap is negligible; they simply reflect the scale of the data.
>
> The task in Figure 4 involves approximating a large teacher MLP with random weights. This is a very challenging task where error reduction is notoriously slow with respect to the capacity of student models, regardless of whether the architecture is an MLP or lmKAN.
>
> Given the factors above, relative efficiency is the most appropriate metric. As illustrated in Figure 4 of the revised manuscript, lmKAN nearly matches the performance of an MLP with a $4\times$ larger hidden dimension, which is, thus,  approximately $4^2=16\times$ larger.
> We believe this demonstrates a substantial improvement in expressivity.
>
> Additionally, note that both the absolute and relative improvement is much larger for the methane dataset - lmKANs feature up to 5 times smaller RMSE compared to MLPs with the same hidden dimension, and lmKAN with the hidden dimension of 128 is performing better than MLP with the hidden dimension of 2048, which is approximately $(2048/128)^2=256$ times larger.
>
> > **Weakness 2**
>
> Moradzadeh et al. do compare the accuracy of UKAN against MLPs, but they do not compare the computational efficiency of UKAN against MLPs (they do it only against existing KAN variants), and do not claim that their construction is Pareto optimal against MLPs, which is relevant to the discussion in the related work of our manuscript.
>
> In the revised version of our manuscript, we phrase this more accurately.
>
> > **Weakness 3**
>
> The grid size behaves like any standard architectural hyperparameter (e.g., width or depth, learning rate, weight decay, and many others) and does not impose a unique optimization burden. Its impact on the accuracy typically follows a predictable U-shaped curve, as Figures 4, 9, 24, and 26  of the revised manuscript illustrate. Thus, it makes it straightforward to tune rather than requiring an exhaustive random search. Furthermore, one can use efficient tuning strategies identical to those used for standard MLPs, such as finding the optimal value for a small model, instead of doing an expensive search for a large one.
>
> > **Question 1**
>
> Overall, our manuscript is focused on the inference cost (e.g., we explicitly mention only inference cost in the abstract). During training, we converged all the models very tightly (as demonstrated by the 1/2 lines for MLPs). The thorough systematic analysis of the training cost would require many more numerical experiments and much more space.
>
> Nevertheless, as a preliminary experiment, we tried to fit lmKANs on the methane dataset for as little as 1 hour on the RTX 3080 GPU (as opposed to H100), and found that it is already sufficient to achieve Pareto optimal performance (accuracy vs inference cost) compared to fully converged MLP models.
>
> > **Question 2**
>
> Figure 4 of the revised manuscript presents the analysis of the performance of the lmKAN models as a function of the number of trainable parameters per inner function ($(G+1)^2$ for the 2D lmKANs).
> Additionally, the similar plots for other setups are given in Figures 9, 24, and 26. Overall, in all of the cases, the performance of lmKANs behaves as a U-shaped curve, thus making the search for optimal G straightforward. For "how much increasing the parameter capacity of KAN's improves expressivity," see also the discussion for Weakness 1.
>
> ---
>
> We thank the reviewer again for their detailed feedback and look forward to their reply.

---

### Meta-Review · Area_Chair_JVYf · 2026-01-12

**Summary:**

All reviewers suggest acceptance except of 63pm. I summarize the concerns of reviewer 63pm below:

> The paper claims that lmKANs can add parameters without increasing inference cost by increasing grid resolution. This claim is not tested. The main experiments vary network width, just like MLPs, so they do not evaluate the stated advantage of grid scaling. The key experiment is missing. Network width should be fixed while grid resolution is increased, and both performance and inference cost should be reported. Without this, the main claim is unsupported. Grid resolution results appear only in the appendix and are incomplete. They show performance but not inference cost. Some results show performance getting worse for large grids, which contradicts the paper’s narrative and is not discussed. The motivation for moving from 1D to 2D models is based on an unfair comparison. The baseline uses Gaussian RBFs rather than splines, so the instability cannot be attributed to 1D grids. A proper 1D spline baseline is missing.

**Reviewer Concerns:**

The reviewers seem to have addressed the above concerns.

**Reviewer Scores:**

The majority of them suggest to accept the paper. So I guess, this would have been their final suggestion as well.

---

### Decision · Program_Chairs · 2026-01-26

Accept (Poster)